# LIPSUM-FT: ROBUST FINE-TUNING OF ZERO-SHOT MODELS USING RANDOM TEXT GUIDANCE

**Giung Nam**[1]    **Byeongho Heo**[2]    **Juho Lee**[1,3]
[1]KAIST AI    [2]NAVER AI Lab    [3]AITRICS
{giung, juholee}@kaist.ac.kr,   bh.heo@navercorp.com

## ABSTRACT

Large-scale contrastive vision-language pre-trained models provide the zero-shot model achieving competitive performance across a range of image classification tasks without requiring training on downstream data. Recent works have confirmed that while additional fine-tuning of the zero-shot model on the reference data results in enhanced downstream performance, it compromises the model's robustness against distribution shifts. Our investigation begins by examining the conditions required to achieve the goals of *robust fine-tuning*, employing descriptions based on feature distortion theory and joint energy-based models. Subsequently, we propose a novel robust fine-tuning algorithm, Lipsum-FT, that effectively utilizes the language modeling aspect of the vision-language pre-trained models. Extensive experiments conducted on distribution shift scenarios in DomainNet and ImageNet confirm the superiority of our proposed Lipsum-FT approach over existing robust fine-tuning methods.

## 1 INTRODUCTION

Recent advances in visual representation learning have been achieved through large-scale contrastive vision-language pre-training (Radford et al., 2021; Jia et al., 2021; Pham et al., 2023). A prominent instance that exemplifies this trend is the Contrastive Language-Image Pre-training (CLIP; Radford et al., 2021), a methodology that leverages natural language supervision to enhance visual representation learning. Capitalizing on its language modeling component, the zero-shot CLIP model, in conjunction with a classification head tailored using class label text for downstream tasks, adeptly conducts image classification tasks without requiring extra optimization specifically on the downstream images.

Although the vision-language models achieve remarkable performance without extra training in downstream tasks, fine-tuning on the reference data still proves to be an effective way to improve downstream performance significantly. Nonetheless, this fine-tuning procedure compromises robustness: the accuracy of the fine-tuned model decreases across distribution shifts compared to the accuracy of the initial zero-shot model (Radford et al., 2021; Pham et al., 2023). Thus, it is worth exploring a novel technique for *robust fine-tuning* that can alleviate the performance trade-off between reference and distribution shift data.

While recent works have been conducted in developing robust fine-tuning methods applicable to vision-language models (Wortsman et al., 2022b; Tian et al., 2023; Mao et al., 2023), these approaches tend to neglect the incorporation of the language aspect during the fine-tuning process. Since the nature of vision-language models is their multimodal structure encompassing both images and text, we argue that the existing robust fine-tuning methods do not effectively harness the information embedded within the language modeling aspect. Consequently, we propose a novel strategy for robust fine-tuning that effectively utilizes the language model, aiming to enhance robustness.

The main contributions of our work can be outlined as follows:

- We re-verified the performance trade-off on the reference and distribution shift data after fine-tuning of CLIP-ViT models (Radford et al., 2021). Our empirical investigation revealed that the recently introduced feature distortion theory (Kumar et al., 2022) lacks

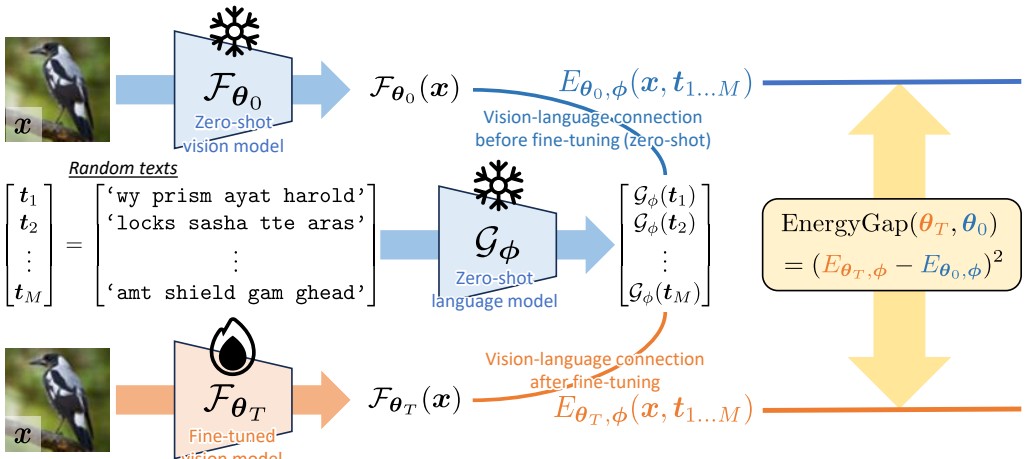

**Figure 1: Overview.** We first verify that standard fine-tuning declines the vision-language connections in the pre-trained CLIP model, as evidenced by changes in the energy function $E_{\boldsymbol{\theta},\phi}$ after $T$ fine-tuning steps, denoted as $\boldsymbol{\theta}_0 \to \boldsymbol{\theta}_T$ (§ 4.2). Subsequently, we propose a simple yet effective novel robust fine-tuning method, Lipsum-FT, which regularizes the $\mathrm{EnergyGap}(\boldsymbol{\theta}_T, \boldsymbol{\theta}_0)$ easily derived from language model outputs for *random texts* during fine-tuning (§ 5.1).

a comprehensive explanation for the robustness observed in distribution shifts during the fine-tuning of zero-shot CLIP-ViT models.

- In lieu of the feature distortion theory, we introduced a concept of joint energy-based models (Grathwohl et al., 2019) to investigate the underlying factors contributing to the robustness of the existing robust fine-tuning methods. To summarize, our research showed that the fine-tuning process disturbs the connection between the vision and language models, as evidenced by alterations in the energy values.

- We propose a novel robust fine-tuning method, Lipsum-FT, tailored for vision-language models. Lipsum-FT utilizes language model outputs to align the fine-tuned model with the zero-shot model. Notably, Lipsum-FT is the pioneering effort that considers the language modeling component of vision-language models for robust fine-tuning.

- Figure 1 depicts an overall idea of our proposed Lipsum-FT approach. Through empirical validation using the DomainNet and ImageNet datasets to simulate distribution shift scenarios, we demonstrate that Lipsum-FT outperforms previous methods in terms of both prediction accuracy and uncertainty estimation (§ 5.2).

## 2 PROBLEM SETTING

Throughout the paper, we consider image classification tasks using pre-trained vision-language models (Radford et al., 2021; Jia et al., 2021; Pham et al., 2023). To elaborate on our problem setting, we will primarily discuss CLIP (Radford et al., 2021), yet it is worth mentioning that other vision-language models can be described in a similar manner.

**Zero-shot image classification.** CLIP consists of two separate models; (1) a vision model $\mathcal{F}_{\boldsymbol{\theta}}$ parameterized by $\boldsymbol{\theta}$, which maps images into the $D$-dimensional representation space, and (2) a language model $\mathcal{G}_{\phi}$ parameterized by $\phi$, which maps texts into the same representation space. For a $K$-class classification problem taking images $\boldsymbol{x}$, CLIP demonstrates the ability to predict the target label $y$ in a zero-shot manner. Specifically, we can compute the class logits as $\boldsymbol{u}(\boldsymbol{x}) = \mathbf{W}\mathcal{F}_{\boldsymbol{\theta}}(\boldsymbol{x})$, where the classification head weights $\mathbf{W}$ is configured by the language model $\mathcal{G}_{\phi}$ and the class label text $\boldsymbol{t}_1, \ldots, \boldsymbol{t}_K$ (e.g., 'a photo of a *class_name*'),

$$\mathbf{W} = [\mathcal{G}_{\phi}(\boldsymbol{t}_1) \quad \cdots \quad \mathcal{G}_{\phi}(\boldsymbol{t}_K)]^{\top} \in \mathbb{R}^{K \times D}. \tag{1}$$

The logits $\boldsymbol{u}(\boldsymbol{x}) \in \mathbb{R}^K$ for input $\boldsymbol{x}$ are transformed into a $K$-class probability vector using the softmax function, denoted as $\boldsymbol{p}(y|\boldsymbol{x}) = \mathrm{Softmax}(\boldsymbol{u}(\boldsymbol{x}))$. Namely,

$$p(y = k|\boldsymbol{x}) = \mathrm{Softmax}^{(k)}(\boldsymbol{u}(\boldsymbol{x})) = \frac{\exp\left(\boldsymbol{u}^{(k)}(\boldsymbol{x})\right)}{\sum_{j=1}^{K} \exp\left(\boldsymbol{u}^{(j)}(\boldsymbol{x})\right)}, \text{ for } k = 1, \ldots, K. \tag{2}$$

**Fine-tuning of the zero-shot model.** The zero-shot CLIP model demonstrates competitive performance in image classification tasks. However, fine-tuning the zero-shot model can further improve the downstream performance of the zero-shot model. A conventional fine-tuning procedure involves simultaneously updating the vision model parameters $\boldsymbol{\theta}$ and the downstream classification head $\mathbf{W}$ to minimize the cross-entropy loss function,

$$\mathcal{L}_{\mathrm{CE}}(\boldsymbol{\theta}, \mathbf{W}) = -\sum_{(\boldsymbol{x},k)\in\mathcal{D}} \log\left(\mathrm{Softmax}^{(k)}(\mathbf{W}\mathcal{F}_{\boldsymbol{\theta}}(\boldsymbol{x}))\right) \tag{3}$$

where $\boldsymbol{x}$ and $k$ respectively are an image and its corresponding label index in the training dataset $\mathcal{D}$. We refer readers to Appendix B.1 for a more detailed description.

## 3 RELATED WORK

This section mainly focuses on the works that follow the advent of the large-scale vision-language model (Radford et al., 2021). Appendix A provides further discussion of previous work in transfer learning, out-of-distribution generalization, and other relevant areas.

**Robust fine-tuning of vision-language models.** In line with the recent progress in large-scale vision-language models, several works have recently delved into the realm of robust fine-tuning of these models: Wortsman et al. (2022b) demonstrated that a simple weight averaging strategy, which ensembles the weights of the zero-shot and fine-tuned models, effectively enhances robustness; Tian et al. (2023) proposed a method where the fine-tuned weights are projected into the neighboring region of the zero-shot model using different projection radii for each layer. In a closely related work to ours, Mao et al. (2023) utilized context information derived from the language model.

**Theory of transfer learning.** While much of the current research on transfer learning focuses primarily on linear probing due to the complexity involved in analyzing the fine-tuning process (Tripuraneni et al., 2020; Du et al., 2021), a recent investigation by Kumar et al. (2022) introduced the concept termed the *feature distortion theory*. This theory proposes that fine-tuning could potentially distort the features learned during pre-training, potentially resulting in decreased performance on data that falls outside the distribution of the training data. However, as real-world scenarios may not always align with theoretical assumptions, the feature distortion theory alone does not provide a comprehensive explanation for robustness in practical applications (Trivedi et al., 2023).

## 4 UNDERSTANDING FINE-TUNING OF ZERO-SHOT MODELS

This section examines the implications of fine-tuning for zero-shot vision-language models on vision tasks, particularly concerning distribution shifts. To this end, we present an extensive array of experimental outcomes across the following two scenarios using CLIP-ViT: **(1) DomainNet**, where `DomainNet-R` is employed as the reference training data, and evaluation encompasses four distribution shifts `DomainNet-{P,C,I,S}`; **(2) ImageNet**, where `ImageNet` serves as the reference training data, and evaluation entails four distribution shifts `ImageNet-{V2,R,A,S}`. Please refer to Appendix C for more details on these datasets.

### 4.1 STANDARD FINE-TUNING

It is common practice to initialize a downstream classification head with zero-shot weights derived from the language model when fine-tuning vision-language models (Wortsman et al., 2022b; Tian et al., 2023; Mao et al., 2023). We denote this standard fine-tuning approach as `FT` throughout the paper and refer readers to Appendix B.1 for supplementary fine-tuning results using alternative head initialization strategies.

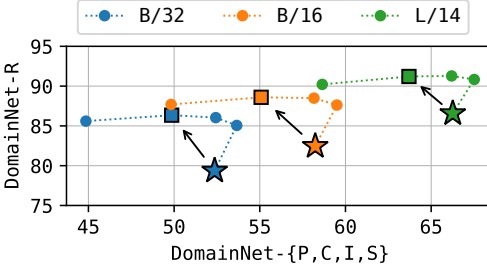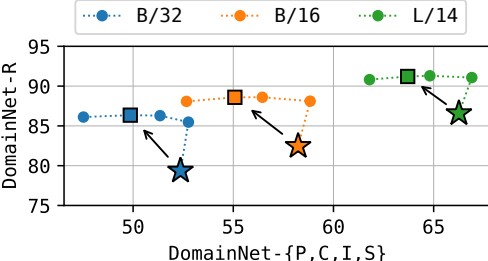

**Figure 2: Trade-off plots on DomainNet.** In plots, the vertical axis shows accuracy on the reference data, whereas the horizontal axis indicates average accuracy on distribution shifts, i.e., the top-right represents a better case. The star markers ☆ correspond to the zero-shot model, while the square markers □ denote the model fine-tuned with `5000` training steps and a learning rate of `1e-05`. **Left:** The number of training steps is kept at `5000`, while the learning rate varies as `1e-06`, `3e-06`, `1e-05`, and `3e-05` (the leftmost point denotes `3e-05`). **Right:** The learning rate is constant at `1e-05`, while the number of training steps is altered to `1000`, `3000`, `5000`, and `10000` (the leftmost point denotes `10000`). Refer to Figure 7 for the results on ImageNet.

**Decrease in performance on distribution shifts.** Radford et al. (2021) observed that fine-tuning CLIP-ViT models leads to improved accuracy on the source distribution, but it also results in a reduction in accuracy on distribution shifts. Figure 2 verifies this trade-off, where more fine-tuning of zero-shot models (represented by star markers ☆) through higher learning rates or more training steps leads to enhanced performance on the reference data (moving upwards), but it also corresponds to a decrease in performance on distribution shifts (moving to the left).

**Examining the feature distortion theory.** We first investigate this phenomenon using the feature distortion theory (Kumar et al., 2022). According to the feature distortion theory, fine-tuning mainly modifies features for the reference data rather than for distribution shift data. Kumar et al. (2022) assert that this is the underlying reason for the performance trade-off observed between reference and distribution shift data and empirically validate it in the context of fine-tuning ResNet-50 that is pre-trained using MoCo v2 (Chen et al., 2020c). They further demonstrate that `LP-FT`, performing fine-tuning after linear-probing, effectively alleviates feature distortion, resulting in enhanced performance on distribution shifts.

However, our findings demonstrate that fine-tuning of CLIP-ViT models does not exhibit greater feature distortion in reference data compared to distribution shift data. In Figure 3, we computed the Euclidean distance between features before and after the fine-tuning procedure to quantitatively measure the extent of distortion (y-axis), in line with the experimental design presented by Kumar et al. (2022). Surprisingly, even in the DomainNet scenario, the features on distribution shift data exhibit larger distortion than those on reference data, contradicting the feature distortion theory. This can be attributed to the notion that the assumptions made in the theory, including 1) the over-parameterized linear network assumption and 2) the rigorous out-of-distribution assumption about the distribution shift data being orthogonal to the reference data's row space, might not align with real-world practical situations. For a more comprehensive understanding of the results and further verification of `LP-FT`, we refer readers to Appendix B.1.

## 4.2    REGULARIZATION TOWARDS ZERO-SHOT FOR ROBUST FINE-TUNING

Differentiating pre-trained vision-language models from vision-only models, e.g., Caron et al. (2021); Chen et al. (2021); Bao et al. (2022); He et al. (2022), is their ability in zero-shot classification utilizing the language model. Notably, the zero-shot model without any training already demonstrates superior performance in handling shifts in data distribution, as illustrated earlier in Figure 2. Consequently, the primary task in achieving robust fine-tuning of vision-language models is maintaining their zero-shot performance on distribution shift data (Wortsman et al., 2022b).

In this section, we delve into a comparative analysis between the existing techniques for achieving robust fine-tuning and the conventional standard fine-tuning approach, denoted as `FT`. To this end,

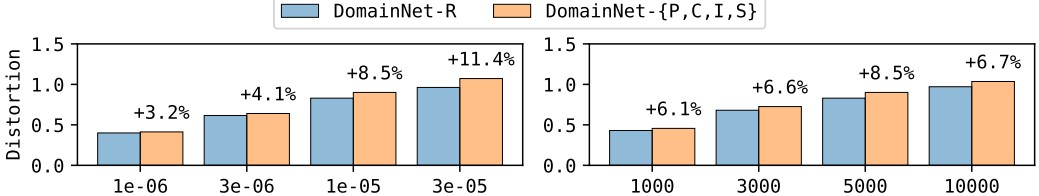

**Figure 3: Bar plots depicting the feature distortion on DomainNet.** A taller bar represents a larger degree of feature distortion after fine-tuning. The number on the bar denotes the relative difference in distortion values between reference and distribution shift data. **Left:** The number of training steps is kept at `5000`, while the learning rate varies as `1e-06`, `3e-06`, `1e-05`, and `3e-05`. **Right:** The learning rate is constant at `1e-05`, while the number of training steps is altered to `1000`, `3000`, `5000`, and `10000`. These plots are with `B/16` on DomainNet, and refer to Figures 9 and 10 for the results with `B/32`, `B/16`, and `L/14` on DomainNet and ImageNet.

we explore the following three groups of existing algorithms that can serve as methods for achieving robust fine-tuning of zero-shot models: (1) `EMA` (Exponential Moving Average) and `L2SP` (Xuhong et al., 2018) apply regularization to steer the fine-tuned solution towards the zero-shot model in the weight space throughout the fine-tuning process; (2) `KD` (Hinton et al., 2015) and `CAR-FT` (Mao et al., 2023) utilize regularization to guide the fine-tuned solution towards the zero-shot model in the output space throughout the fine-tuning procedure; (3) `WiSE` (Wortsman et al., 2022b) and `TPGM` (Tian et al., 2023) are post-hoc approaches that obtain robust solutions by adjusting the fine-tuned model to be positioned close to the zero-shot model in the weight space. All experiments are with `B/16` architecture, and we refer readers to Appendix B.2 for more details on these methods.

**Interpretation via joint energy-based model.** In the pursuit of developing novel robust fine-tuning methods, it becomes essential to consider the following questions: What exactly differentiate the existing robust fine-tuning approaches from standard fine-tuning, and what particular outcomes are realized by employing regularization towards the zero-shot model? To explore this further, we perceive the zero-shot and fine-tuned models through the lens of a joint energy-based model (Grathwohl et al., 2019), which reveals the inherent capacity for generative modeling within conventional discriminative models. To be specific, pre-trained vision-language models offer discriminative modeling for image data $x$ and text data $t$,

$$p_{\theta,\phi}(t|x) = \frac{\exp\left(f_{\theta,\phi}(x,t)\right)}{\sum_{t'}\exp\left(f_{\theta,\phi}(x,t')\right)}, \text{ where } f_{\theta,\phi}(x,t) = \langle \mathcal{F}_{\theta}(x), \mathcal{G}_{\phi}(t)\rangle, \tag{4}$$

and an energy-based model for the joint distribution of $x$ and $t$ can be defined as

$$p_{\theta,\phi}(x,t) = \frac{\exp\left(-E_{\theta,\phi}(x,t)\right)}{Z(\theta,\phi)} = \frac{\exp\left(f_{\theta,\phi}(x,t)\right)}{Z(\theta,\phi)}, \tag{5}$$

where the energy function is given as $E_{\theta,\phi}(x,t) = -f_{\theta,\phi}(x,t)$ and $Z(\theta,\phi)$ is an unknown normalizing constant. That is, the inner product value of $f_{\theta,\phi}(x,t)$ utilized as class logits in discriminative modeling (as described in Eq. 4) can be reused to define the energy function $E_{\theta,\phi}(x,t)$.

Originally, the vision model $\mathcal{F}_{\theta_0}$ before fine-tuning is strongly associated with the language model $\mathcal{G}_{\phi}$ through vision-language contrastive pre-training to the extent that it enables zero-shot discriminative modeling. However, after fine-tuning, the representations of the vision model undergo distortion, resulting in a weakening of the connection between the vision and language models. More precisely, we confirm this through an increase in the *energy gap*, defined as follows:

$$\text{EnergyGap}(\theta,\theta_0) = \mathbb{E}_x\mathbb{E}_t\left(E_{\theta,\phi}(x,t) - E_{\theta_0,\phi}(x,t)\right)^2. \tag{6}$$

Here, we compute the expectation involving $t$ by employing a set of 10,000 text tokens, each having a sequence length of 8. These tokens are generated randomly from the vocabulary of the pre-trained language model. We hypothesize that the existing robust fine-tuning techniques alleviate the decline in pre-trained associations between vision and language models by indirectly reducing the energy gap during their regularization procedure towards the zero-shot model.

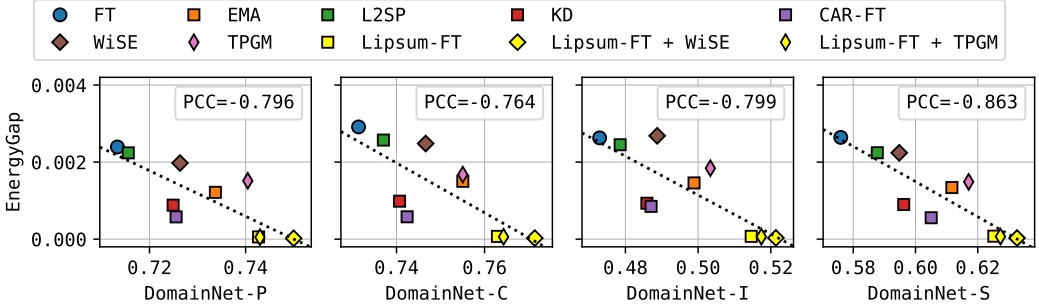

**Figure 4: Energy gaps and distribution shift accuracy on DomainNet.** It represents the energy gap (y-axis) and the relative accuracy of distribution shift data to the reference accuracy (x-axis). The model attained through the fine-tuning procedure is represented by square markers, while the model obtained through post-hoc approaches combining the fine-tuned and zero-shot models is denoted by diamond-shaped markers (i.e., `WiSE` and `TPGM`). The dashed line depicts a linear trend, and it comes with specific details provided as the Pearson correlation coefficient, denoted as `PCC`. We refer readers to Figure 13 for the results on ImageNet.

Figure 4 provides clear evidence for our hypothesis regarding the energy gap minimization aspect in a range of robust fine-tuning approaches. We draw scatter plots illustrating the relative accuracy for distribution shifts (i.e., the distribution shift accuracy divided by the reference accuracy) in connection with the energy gap. The Pearson Correlation Coefficient (PCC) values in each subplot further clarify a distinct correlation between the energy gap and the robustness to distribution shifts. Notably, our method, labeled `Lipsum-FT` and elaborated upon in § 5, effectively leverages this connection and achieves superior performance in addressing distribution shift data.

## 5 LIPSUM-FT: FINE-TUNING WITH RANDOM TEXT GUIDANCE

In this section, we propose a novel method for robust fine-tuning called `Lipsum-FT`, tailored to enhance the robustness of vision-language models after fine-tuning. Briefly speaking, the core concept of our approach is to minimize the change in energy during the fine-tuning process in the context of discriminative modeling with the language model.

### 5.1 LIPSUM-FT

Alongside the cross-entropy loss, which is a main objective of the fine-tuning procedure, our proposed `Lipsum-FT` method also minimizes the following regularization term,

$$\hat{\mathcal{R}}(\boldsymbol{\theta}) = \frac{1}{2M} \left\| \boldsymbol{v}_{\boldsymbol{\theta},\boldsymbol{\phi}}(\boldsymbol{x}) - \boldsymbol{v}_{\boldsymbol{\theta}_0,\boldsymbol{\phi}}(\boldsymbol{x}) \right\|_2^2, \tag{7}$$

where $\boldsymbol{v}_{\boldsymbol{\theta},\boldsymbol{\phi}}(\boldsymbol{x})$ is computed using the language model $\mathcal{G}_{\boldsymbol{\phi}}$ and $M$ text tokens $\boldsymbol{t}_1, \ldots, \boldsymbol{t}_M$,

$$\boldsymbol{v}_{\boldsymbol{\theta},\boldsymbol{\phi}}^{(m)}(\boldsymbol{x}) = \langle \mathcal{G}_{\boldsymbol{\phi}}(\boldsymbol{t}_m), \mathcal{F}_{\boldsymbol{\theta}}(\boldsymbol{x}) \rangle, \text{ for } m = 1, \ldots, M. \tag{8}$$

The name `Lipsum-FT` comes from the process of generating text tokens $\boldsymbol{t}_1, \ldots, \boldsymbol{t}_M$ in a *random* manner using the vocabulary of the pre-trained vision-language model. It is worth mentioning that the minimization of the regularization term described in Equation 7 is a stochastic way to minimize the energy gap defined before in Equation 6, which involves $M$ samples for every fine-tuning iteration. Here, we have defined our regularization term in the *logit matching* (Ba & Caruana, 2014) format to facilitate a comparison with the closely related `CAR-FT`. More detailed discussions regarding our proposed `Lipsum-FT` regularization will be provided in the forthcoming § 5.3.

### 5.2 EVALUATION RESULTS

To begin, we present the evaluation results clearly showing the efficacy of our approach compared to the baseline methods. For a more comprehensive view of the results, please see Appendix B.3.

**Table 1: Main results on distribution shift tasks.** It summarizes the accuracy of various methods for `B/16` in distribution shift scenarios on DomainNet and ImageNet. An underline highlights the top two values in each column, where all values are averaged from five measurements. See Table 5 for the results with `B/32` and `L/14`.

| Method | DomainNet | | | | | ImageNet | | | | |
|---|---|---|---|---|---|---|---|---|---|---|
| | R | P | C | I | S | IN | V2 | R | A | S |
| Zero-shot | 82.5 | 66.7 | 65.6 | 43.8 | 56.9 | 68.2 | 62.0 | 76.3 | 52.1 | 46.7 |
| FT | 88.5±0.1 | 62.5±0.2 | 64.3±0.4 | 41.2±0.5 | 50.6±0.5 | 82.8±0.1 | 72.6±0.3 | 68.5±0.3 | 39.2±0.3 | 48.0±0.2 |
| EMA | 88.9±0.0 | 65.5±0.4 | 67.3±0.6 | 44.4±0.6 | 54.8±0.8 | 83.0±0.0 | 72.6±0.2 | 70.2±0.3 | 39.6±0.3 | 49.4±0.3 |
| L2SP | 88.6±0.1 | 63.2±0.2 | 65.0±0.2 | 42.0±0.2 | 51.4±0.7 | 82.9±0.1 | 72.6±0.2 | 68.8±0.2 | 39.7±0.2 | 48.2±0.1 |
| KD | 88.6±0.1 | 64.2±0.2 | 65.7±0.3 | 42.9±0.2 | 52.4±0.4 | 83.1±0.1 | 73.1±0.3 | 72.9±0.1 | 42.3±0.4 | 49.9±0.2 |
| CAR-FT | 88.9±0.0 | 64.4±0.1 | 65.8±0.2 | 43.3±0.2 | 53.2±0.5 | 83.2±0.0 | 73.0±0.2 | 71.3±0.3 | 43.7±0.2 | 49.5±0.2 |
| Lipsum-FT | 89.0±0.0 | 66.3±0.2 | 68.0±0.0 | 46.0±0.2 | 56.2±0.2 | 83.3±0.0 | 73.6±0.1 | 75.9±0.1 | 49.9±0.3 | 51.4±0.1 |

**Table 2: Uncertainty quantification results on distribution shift tasks.** It summarizes the uncertainty metrics, including (a) expected calibration error and (b) negative log-likelihood, of various methods for `B/16` in distribution shift scenarios on DomainNet and ImageNet. All fine-tuning results are averaged from five measurements, and the corresponding standard deviation values are provided. An underline highlights the top two values in each column.

**(a)** Expected calibration error (lower is better).

| Method | DomainNet | | | | | ImageNet | | | | |
|---|---|---|---|---|---|---|---|---|---|---|
| | R | P | C | I | S | IN | V2 | R | A | S |
| Zero-shot | 0.025 | 0.020 | 0.028 | 0.107 | 0.021 | 0.019 | 0.023 | 0.039 | 0.076 | 0.045 |
| FT | 0.024±0.001 | 0.123±0.001 | 0.122±0.004 | 0.204±0.004 | 0.184±0.004 | 0.034±0.001 | 0.075±0.001 | 0.066±0.002 | 0.234±0.002 | 0.159±0.002 |
| EMA | 0.018±0.002 | 0.104±0.005 | 0.101±0.009 | 0.196±0.006 | 0.151±0.011 | 0.027±0.002 | 0.068±0.003 | 0.050±0.003 | 0.220±0.004 | 0.142±0.004 |
| L2SP | 0.024±0.001 | 0.123±0.002 | 0.124±0.002 | 0.204±0.003 | 0.183±0.004 | 0.033±0.001 | 0.074±0.002 | 0.063±0.001 | 0.228±0.002 | 0.158±0.002 |
| KD | 0.006±0.001 | 0.078±0.003 | 0.073±0.003 | 0.170±0.005 | 0.123±0.002 | 0.007±0.001 | 0.029±0.002 | 0.009±0.001 | 0.169±0.004 | 0.083±0.002 |
| CAR-FT | 0.018±0.001 | 0.110±0.001 | 0.107±0.001 | 0.202±0.004 | 0.156±0.003 | 0.026±0.001 | 0.066±0.002 | 0.045±0.001 | 0.197±0.003 | 0.141±0.001 |
| Lipsum-FT | 0.007±0.000 | 0.055±0.002 | 0.043±0.001 | 0.146±0.001 | 0.081±0.001 | 0.008±0.001 | 0.034±0.001 | 0.012±0.002 | 0.122±0.004 | 0.088±0.001 |

**(b)** Negative log-likelihood (lower is better).

| Method | DomainNet | | | | | ImageNet | | | | |
|---|---|---|---|---|---|---|---|---|---|---|
| | R | P | C | I | S | IN | V2 | R | A | S |
| Zero-shot | 0.731 | 1.497 | 1.527 | 3.071 | 2.047 | 1.184 | 1.490 | 0.927 | 1.925 | 2.245 |
| FT | 0.445±0.003 | 1.842±0.010 | 1.748±0.022 | 3.641±0.033 | 2.818±0.041 | 0.624±0.002 | 1.129±0.002 | 1.439±0.015 | 2.765±0.017 | 2.550±0.009 |
| EMA | 0.425±0.003 | 1.673±0.040 | 1.540±0.049 | 3.435±0.068 | 2.469±0.085 | 0.608±0.002 | 1.089±0.009 | 1.330±0.020 | 2.660±0.018 | 2.399±0.025 |
| L2SP | 0.447±0.002 | 1.846±0.007 | 1.736±0.013 | 3.635±0.010 | 2.806±0.043 | 0.621±0.002 | 1.126±0.007 | 1.429±0.010 | 2.727±0.012 | 2.546±0.019 |
| KD | 0.435±0.002 | 1.683±0.010 | 1.575±0.007 | 3.427±0.023 | 2.511±0.030 | 0.601±0.002 | 1.047±0.002 | 1.137±0.007 | 2.407±0.019 | 2.257±0.022 |
| CAR-FT | 0.423±0.001 | 1.732±0.005 | 1.624±0.013 | 3.505±0.024 | 2.579±0.035 | 0.599±0.001 | 1.087±0.005 | 1.261±0.011 | 2.424±0.014 | 2.381±0.018 |
| Lipsum-FT | 0.425±0.001 | 1.521±0.005 | 1.400±0.005 | 3.079±0.011 | 2.150±0.011 | 0.595±0.001 | 1.010±0.003 | 0.973±0.004 | 1.986±0.010 | 2.123±0.004 |

**Evaluation results for classification accuracy.** Table 1 presents comparative results in the DomainNet and ImageNet distribution shift scenarios for `B/16`. It clearly shows that `Lipsum-FT` surpasses all other fine-tuning baselines across all types of shifts. Table 5 in Appendix B.3 provides additional confirmation that this excellence extends beyond the `B/16` architecture and it also observed in the case of the `B/32` and `L/14` architectures.

**Evaluation results for predictive uncertainty.** While classification accuracy serves as the primary metric for assessing image classification models, it does not offer insights into the model's ability in uncertainty quantification when processing distribution shift data. Considering the tendency of classification models to make *incorrect* predictions when faced with distribution shift data (as demonstrated by lower distribution shift accuracy compared to reference accuracy), evaluating the model's capacity to quantify predictive uncertainty becomes a notable concern.

To account for this, we also present the evaluation results using the following two prominent uncertainty metrics (Guo et al., 2017): (a) Expected calibration error (Pakdaman Naeini et al., 2015), which quantifies calibration by calculating the overall disparity between prediction accuracy and confidence. (b) Negative log-likelihood, which directly assesses the likelihood of predictive cate-

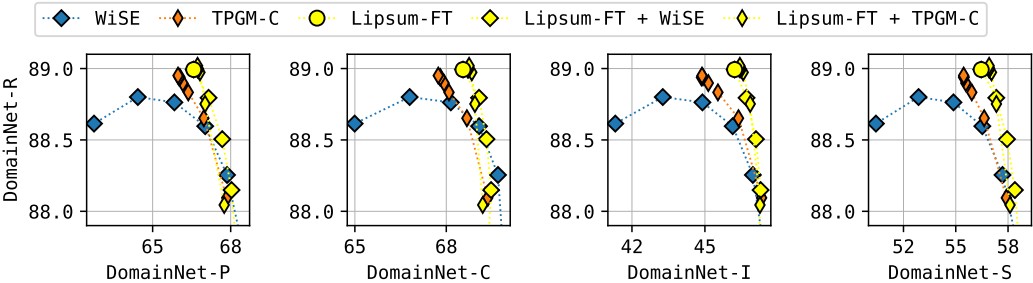

Figure 5: **Scatter plots for post-hoc methods on DomainNet.** Moving in an upward and rightward direction signifies improved accuracy for the reference and distribution shift data, respectively, making the top right corner more desirable. We refer readers to Figure 14 for ImageNet results, as well as Tables 6 and 7 for numerical results.

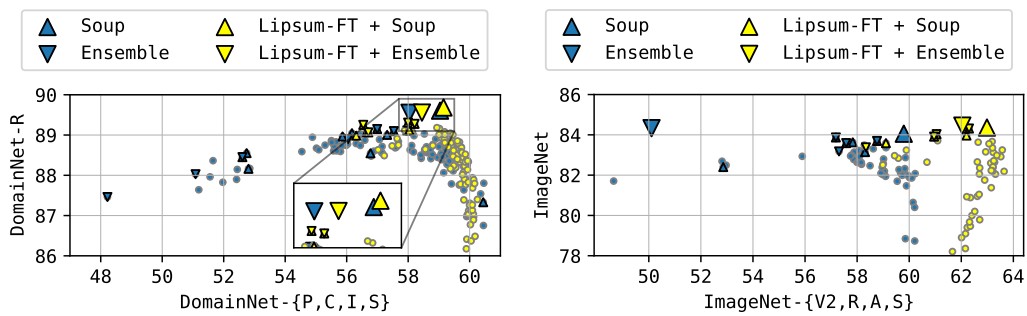

Figure 6: **Scatter plots for ensemble and soup methods.** Moving in an upward and rightward direction signifies improved accuracy for the reference and distribution shift data, respectively, making the top right corner more desirable. Small markers depict a set of fine-tuned models with varying hyperparameter configurations. Within this collection, `Soup` ingredients are identified by a triangular marker, while `Ensemble` members are distinguished by an inverted triangular marker.

gorical distribution. Table 2 shows that our suggested `Lipsum-FT` approach attains lower numbers in these metrics, indicating its superiority in uncertainty quantification.

### 5.3 DISCUSSIONS

**Combining with post-hoc methods.** Given that the `WiSE` and `TPGM` techniques belong to the class of post-hoc methods, which involves merging zero-shot and fine-tuned models, they have the potential to work in conjunction with our proposed `Lipsum-FT`. Accordingly, we further present the results for comparison and combination of `Lipsum-FT` with `WiSE` and `TPGM-C` in Figure 5; `TPGM-C` is a variant of `TPGM` introduced by Tian et al. (2023) for a fair comparison with `WiSE`. Please refer to Appendix B.2 for more details on these methods.

Figure 5 clearly shows that `Lipsum-FT` is positioned in a favorable region at the upper right, demonstrating its superiority in both the reference data (upper) and distribution shift data (right). Furthermore, the plots labeled as `Lipsum-FT + WiSE` and `Lipsum-FT + TPGM-C` illustrate that the existing post-hoc methodologies `WiSE` and `TPGM` can operate independently to improve the distribution shift performance in conjunction with our newly introduced method, `Lipsum-FT`.

**Combining with ensemble and soup methods.** Another strategy acknowledged for its ability to withstand distribution shifts is the concept of *model soup* (Wortsman et al., 2022a), where the weights of multiple models fine-tuned with different hyperparameter configurations are averaged. In this context, we also present the results comparing and combining `Lipsum-FT` with `Soup` and `Ensemble` in Figure 6. `Soup` and `Ensemble` represent weight-space and output-space ensembles, respectively, using the following greedy procedure described in Wortsman et al. (2022a): (1)

**Table 3: Ablation results on loss function and text guidance.** It summarizes the accuracy of methods located within the range from `CAR-FT` to `Lipsum-FT`. The values enclosed in brackets indicate the difference in performance when compared to `CAR-FT`, where all values are averaged from five measurements. An underline highlights the top two values in each column. Please refer to Table 9 for ImageNet results, including standard deviations.

| Method | Loss function $KLD \rightarrow MSE$ | Text guidance $fixed \rightarrow random$ | DomainNet R | P | C | I | S |
|---|:---:|:---:|---|---|---|---|---|
| `CAR-FT` | | | 88.9 | 64.4 | 65.8 | 43.3 | 53.2 |
| `CAR-FT`MSE | ✓ | | 88.9 | 65.7 (+1.3) | 67.1 (+1.3) | 44.5 (+1.2) | 55.2 (+2.0) |
| `Lipsum-FT`KLD | | ✓ | 89.0 (+0.1) | 66.0 (+1.6) | 67.7 (+1.9) | 45.8 (+2.5) | 55.7 (+2.5) |
| `Lipsum-FT` | ✓ | ✓ | 89.0 (+0.1) | 66.3 (+1.9) | 68.0 (+2.2) | 46.0 (+2.7) | 56.2 (+3.0) |

arranging the fine-tuned solutions in order of their validation accuracy, and (2) gradually expanding the pool of candidates for ensembling in a greedy manner to improve the validation accuracy.

Figure 6 illustrates that when our approach is combined with the model soup method, denoted as `Lipsum-FT + Soup`, it results in improvements in both reference and distribution shift performance, positioning it favorably in the upper right region. It is worth emphasizing that the soup models outperform the ensemble models, which come with an extra inference cost (approximately four times more, as we constrained the maximum ensemble size to four). These results align with the observations reported by Wortsman et al. (2022a) and demonstrate that our `Lipsum-FT` approach functions independently from the soup method.

**Further comparison with Mao et al. (2023).** In a related study conducted by Mao et al. (2023), which introduced the `CAR-FT` method, they empirically investigated the impact of fine-tuning on the context awareness of pre-trained CLIP features. Their findings align with our central idea that fine-tuning disrupts the pre-trained connection between vision and language models. However, our research extends beyond context awareness, encompassing a broader spectrum of general relationships for all potential discriminative models derived from pre-trained vision-language models. We enhance our understanding of the critical factors behind `Lipsum-FT`'s success by conducting additional ablation study involving a comparative analysis between `CAR-FT` and `Lipsum-FT`.

More precisely, the points mentioned above lead to the following methodological distinctions between `CAR-FT` and our proposed `Lipsum-FT`: (1) `CAR-FT` utilizes the *Kullback-Leibler Divergence (KLD)* loss for handling categorical outputs in the context of a discriminative model for context classification. In contrast, `Lipsum-FT` adopts the *Mean Squared Error (MSE)* loss for energies (i.e., logits) in the context of an arbitrary discriminative model. (2) `CAR-FT` employs a *fixed* text guidance to explicitly address context awareness, whereas `Lipsum-FT` relies on a *random* text guidance strategy covering all potential aspects inherent in pre-trained vision-language models. Table 3 conducts an ablation analysis on these two factors, illustrating the roles played by both the MSE loss and the random text guidance strategy in enhancing performance. For a more detailed implementation and a connection between MSE and KLD losses, refer to Appendix B.3.

## 6 CONCLUSION

In this study, we introduced `Lipsum-FT`, a straightforward yet powerful method for enhancing the distribution shift robustness of fine-tuned vision-language models. `Lipsum-FT` goes beyond the conventional notion that fine-tuning distorts the features learned during pre-training; it advances the concept in the context of pre-trained vision-language models that fine-tuning can disrupt the pre-trained connections between vision and language models. From this, `Lipsum-FT` minimizes the energy gap across all potential discriminative models derived from the pre-trained vision-language model. The extensive experimental results strongly support these claims, and `Lipsum-FT` has also demonstrated its capability to enhance other methods using weight averaging strategies. This research represents pioneering efforts in involving the language model aspect into the fine-tuning procedure for vision-language models. An intriguing avenue for future research is investigating a novel text guidance strategy tailored for robust fine-tuning instead of random text guidance.

**Ethics statement.** In this work, we utilized CLIP, a notable contemporary example of large-scale pre-trained models. Such large models potentially raise ethical concerns due to unfiltered data during their pre-training phase (Weidinger et al., 2021). Nevertheless, it is worth clarifying that this work is fundamentally an analytical paper and does not inherently carry significant ethical risks.

**Reproducibility statement.** Appendix C provides specific information about the experiments, and we plan to release the experiment code accessible to the public to facilitate reproducibility.

## ACKNOWLEDGEMENT

This work was partly supported by Institute of Information & communications Technology Promotion(IITP) grant funded by the Korea government(MSIT)(No.2019-0-00075, Artificial Intelligence Graduate School Program(KAIST)), the National Research Foundation of Korea(NRF) grant funded by the Korea government(MSIT) (NRF-2022R1A5A708390812, NRF-2021M3E5D9025030), and KAIST-NAVER Hypercreative AI Center. This material is based upon work supported by the Google Cloud Research Credits program with the award GCP19980904 and Cloud TPUs from Google's TPU Research Cloud (TRC).

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

## A    ADDITIONAL RELATED WORK

**Transfer learning.** The notion that training convolutional deep neural networks on extensive upstream data leads to obtaining generalizable feature representations (Zeiler & Fergus, 2014; Yosinski et al., 2014; Oquab et al., 2014) has established the foundation for the *pre-training-and-fine-tuning* regime in computer vision. Numerous works have proposed different pre-training strategies for vision models (Sun et al., 2017; Mahajan et al., 2018; He et al., 2020; Chen et al., 2020a;c;b; Grill et al., 2020; Caron et al., 2020; 2021; Chen et al., 2021; Bao et al., 2022; He et al., 2022), enabling effective adaptation of pre-trained knowledge to downstream vision tasks. This adaptation process typically involves fine-tuning, and various methods for regularizing towards the initial pre-trained model have also emerged to preserve the pre-trained knowledge during the fine-tuning process (Xuhong et al., 2018; Li et al., 2019; 2020; Zhang et al., 2020; Gouk et al., 2021). The advent of pre-trained vision-language models capable of zero-shot classification (Radford et al., 2021; Jia et al., 2021; Pham et al., 2023) has triggered the development of this regularization concept towards transferring the distribution shift robustness from the zero-shot model to the fine-tuned model (Wortsman et al., 2022b; Mao et al., 2023; Tian et al., 2023).

**Energy-based model.** An energy-based model belongs to the category of generative models, which gain an understanding of an underlying data distribution by examining a sample dataset (Teh et al., 2003). One notable characteristic of energy-based models is their ability to assess the likelihood that a given set of data originates from a specific data distribution (LeCun et al., 2006). The seminal work of Xie et al. (2016) demonstrated that there is a mutual derivation between the discriminative model and the energy-based model. Following this, Grathwohl et al. (2019) suggested a reinterpretation of a conventional discriminative classifier as an energy-based model for the joint distribution.

**Handling out-of-distribution data.** Given the inability to handle every potential instance during the training phase, mainly due to the *dataset shift* (Quinonero-Candela et al., 2008), accounting for unknowns in testing becomes a notable issue in machine learning (Scheirer et al., 2013; Bendale & Boult, 2015; 2016). Within this context, the following problems have emerged as a significant research topic in machine learning:

- *Out-of-distribution detection* problem aims to answer the following (Hendrycks & Gimpel, 2017): *can we identify whether a test sample originates from a distribution distinct from the one employed during the training process?* As an answer, one way to quantify the probability that given data came from a specific data distribution is through the notion of *energy* (LeCun et al., 2006), where increased energy levels suggest that the data points exhibit lower probability density (Grathwohl et al., 2019; Liu et al., 2020). Notably, a recent work by Ming et al. (2022) demonstrated the effective use of CLIP-like models for zero-shot out-of-distribution detection, capitalizing on the rich pre-trained feature representations obtained through a multi-modal contrastive loss.

- *Out-of-distribution generalization* problem explores *whether a model can effectively handle a test sample that shows specific variations from the training dataset while still being part of a comparable overall distribution*. From the perspective of *domain shift*, assuming an underlying latent representation of the covariate space that is unchanged (Quinonero-Candela et al., 2008), there are closely related research topics such as *domain adaptation* and *domain generalization*. However, the out-of-distribution generalization encompasses a broader scope than domain adaptation, which assumes access to the test data distribution, or domain generalization, which involves utilizing data from diverse domains (Muandet et al., 2013; Blanchard et al., 2011). Especially in computer vision, many studies have approached the out-of-distribution generalization problem by focusing on the distribution shift robustness of image classification models; models trained on reference data should provide reliable predictions even when dealing with shifted data (Torralba & Efros, 2011; Recht et al., 2019; Taori et al., 2020).

**Prompt learning.** The seminal work of Radford et al. (2021) has already demonstrated the significance of crafting effective prompts for enhancing zero-shot classification with CLIP. Utilizing suitable prompt text through *prompt engineering* improves zero-shot classification performance compared to the baseline of using context-free class text. Consequently, a branch of *prompt learning* research has arisen, focusing on adapting pre-trained vision-language models to downstream vision tasks *without fine-tuning vision model parameters* (Zhou et al., 2022b;a; Du et al., 2022; Jia et al.,

2022; Shu et al., 2022). Empirical findings from prior research have validated that prompt learning methods exhibit competitive performance in out-of-distribution generalization, including the ImageNet scenario we discussed in this paper (Zhou et al., 2022b;a; Shu et al., 2022). However, their reference performance is considerably lower than the fine-tuning techniques we considered, with the highest accuracy in Shu et al. (2022) reaching 73.6, while our `FT` baseline achieves 82.8.

# B    SUPPLEMENTARY MATERIALS

## B.1    STANDARD FINE-TUNING

**Details on standard fine-tuning procedure.** Starting from an initial parameters $\boldsymbol{\Theta}_0 = (\boldsymbol{\theta}_0, \mathbf{W}_0)$, a standard fine-tuning procedure obtains the fine-tuned model parameters $\boldsymbol{\Theta}_T = (\boldsymbol{\theta}_T, \mathbf{W}_T)$ after $T$ iterations. Specifically, the update rule for the $t^{\text{th}}$ iteration is

$$\boldsymbol{\Theta}_t = \boldsymbol{\Theta}_{t-1} - \eta \cdot \nabla_{\boldsymbol{\Theta}} \hat{\mathcal{L}}_{\text{CE}}(\boldsymbol{\Theta})|_{\boldsymbol{\Theta} = \boldsymbol{\Theta}_{t-1}}, \text{ for } t = 1, \dots, T, \tag{9}$$

where $\eta$ is a learning rate and $\hat{\mathcal{L}}_{\text{CE}}(\boldsymbol{\theta})$ is a cross-entropy loss computed for a given mini-batch. Although we practically employed the Adam optimizer (Kingma & Ba, 2015) and a cosine decay learning rate scheduler with linear warm-up, we present here the basic form for stochastic gradient methods (Robbins & Monro, 1951) for the sake of clarity. Unless specified, the mini-batch size was set to `256`, the peak learning rate was configured as `1e-05` with the initial 10% of steps were dedicated to the warm-up phase. We also evaluate fine-tuned models every `1000` iterations and choose the model that demonstrates the highest performance on the reference validation data, which corresponds to `DomainNet-R` for the DomainNet scenario and `ImageNet` for the ImageNet scenario.

**Head initialization matters.** The feature distortion theory of Kumar et al. (2022) underscores how the fine-tuning procedure is influenced by the initial weights of the head. Consequently, we explore three distinct strategies for initializing the head during the standard fine-tuning of CLIP-ViT models: (1) `Scratch-FT`: fine-tuning with head weights initialized to zero; (2) `LP-FT`: fine-tuning with linear-probed head weights; (3) `FT`: fine-tuning with zero-shot head weights. We obtain the zero-shot head weights by following the procedure described in the official code base.[1]

Table 4 summarizes the baseline results for standard fine-tuning with three different head initializations. Across the board, `FT` consistently exhibits superior performance on distribution shifts compared to `Scratch-FT` and `LP-FT` on average. It highlights the considerable influence exerted by the distinctive aspect of zero-shot head initialization involving the language module in the zero-shot vision-language model fine-tuning scenario.

**Supplementary plots for Figure 2.** In Figure 7, we present the same plot for the ImageNet scenario. As discussed in § 4.1, the performance on distribution shifts deteriorates as we engage in further fine-tuning via increased learning rates or additional training steps. We also provide detailed results for each distribution shift in Figure 8.

**Supplementary plots for Figure 3.** In Figures 9 and 10, we present outcomes from parallel experiments for `B/32`, `B/16`, and `L/14` across both DomainNet and ImageNet scenarios. As discussed in § 4.1, there is no significant difference in the extent of the feature distortion between reference and distribution shift data, and even in the case of DomainNet, the feature distortion for distribution shift data is larger than that for reference data. Figure 8 also provides detailed results for each distribution shift (bottom plots in each subfigure).

**Further examination on the feature distortion theory.** Continuing from § 4.1, we delve into an empirical exploration to ascertain whether the feature distortion theory can elucidate the scenario of CLIP-ViT fine-tuning. Across `B/32`, `B/16`, and `L/14` architectures on DomainNet and ImageNet datasets, Figures 11 and 12 demonstrate: (1) `LP-FT` effectively reduces feature distortion, nonetheless, (2) there exists no noteworthy discrepancy in the degree of distortion between reference and distribution shift data on average. Considering the fact that `LP-FT`'s performance on distribution shifts is not superior to other methods, as evident in Table 4, we conclude that `LP-FT` does not achieve the goal of robust fine-tuning in the scenario of CLIP-ViT fine-tuning.

---

[1]https://github.com/openai/CLIP

**Table 4: Baseline results for standard fine-tuning.** Classification accuracy for both reference and distribution shift data. The number of training steps is `5000` for DomainNet and `50000` for ImageNet, using a learning rate of `1e-05`. The highest value across fine-tuning methods is indicated with an underline. Moreover, we provide the results for zero-shot ('Zero-shot') and linear-probed models ('Linear-probed') for further comparisons.

**(a)** DomainNet.

| Model | Method | R | Distribution shifts | | | | |
| | | | P | C | I | S | AVG |
|---|---|---|---|---|---|---|---|
| B/32 | Scratch-FT | $85.0_{\pm0.1}$ | $55.8_{\pm0.2}$ | $57.2_{\pm0.3}$ | $32.1_{\pm0.2}$ | $43.9_{\pm0.2}$ | 47.3 |
| | LP-FT | $84.4_{\pm0.0}$ | $50.5_{\pm0.3}$ | $51.4_{\pm0.3}$ | $27.2_{\pm0.2}$ | $36.5_{\pm0.3}$ | 41.4 |
| | FT | $\underline{86.5}_{\pm0.1}$ | $\underline{58.9}_{\pm0.2}$ | $\underline{59.7}_{\pm0.3}$ | $\underline{34.8}_{\pm0.2}$ | $\underline{45.9}_{\pm0.3}$ | $\underline{49.8}$ |
| | Linear-probed | 83.2 | 51.4 | 50.7 | 27.2 | 38.6 | 42.0 |
| | Zero-shot | 79.4 | 62.4 | 59.8 | 37.1 | 50.3 | 52.4 |
| B/16 | Scratch-FT | $87.7_{\pm0.0}$ | $60.8_{\pm0.2}$ | $62.7_{\pm0.3}$ | $39.3_{\pm0.2}$ | $49.2_{\pm0.5}$ | 53.0 |
| | LP-FT | $86.7_{\pm0.0}$ | $54.5_{\pm0.1}$ | $56.2_{\pm0.2}$ | $32.3_{\pm0.5}$ | $40.8_{\pm0.6}$ | 46.0 |
| | FT | $\underline{88.5}_{\pm0.1}$ | $\underline{62.5}_{\pm0.2}$ | $\underline{64.3}_{\pm0.4}$ | $\underline{41.2}_{\pm0.5}$ | $\underline{50.6}_{\pm0.5}$ | $\underline{54.7}$ |
| | Linear-probed | 86.0 | 56.0 | 57.4 | 32.3 | 44.2 | 47.5 |
| | Zero-shot | 82.5 | 66.7 | 65.6 | 43.8 | 56.9 | 58.3 |
| L/14 | Scratch-FT | $91.0_{\pm0.1}$ | $67.4_{\pm0.3}$ | $73.9_{\pm0.3}$ | $46.4_{\pm0.3}$ | $62.3_{\pm0.4}$ | 62.5 |
| | LP-FT | $89.3_{\pm0.2}$ | $59.6_{\pm0.5}$ | $66.7_{\pm0.5}$ | $39.5_{\pm0.3}$ | $53.3_{\pm0.4}$ | 54.8 |
| | FT | $\underline{91.2}_{\pm0.1}$ | $\underline{68.1}_{\pm0.2}$ | $\underline{74.3}_{\pm0.4}$ | $\underline{48.6}_{\pm0.6}$ | $\underline{63.5}_{\pm0.5}$ | $\underline{63.6}$ |
| | Linear-probed | 88.8 | 60.5 | 68.0 | 39.1 | 56.1 | 55.9 |
| | Zero-shot | 86.6 | 71.3 | 75.7 | 49.3 | 68.8 | 66.3 |

**(b)** ImageNet.

| Model | Method | IN | Distribution shifts | | | | |
| | | | V2 | R | A | S | AVG |
|---|---|---|---|---|---|---|---|
| B/32 | Scratch-FT | $79.0_{\pm0.1}$ | $\underline{67.2}_{\pm0.3}$ | $55.1_{\pm0.3}$ | $19.1_{\pm0.3}$ | $40.8_{\pm0.1}$ | 45.6 |
| | LP-FT | $78.9_{\pm0.1}$ | $66.5_{\pm0.1}$ | $54.4_{\pm0.2}$ | $19.0_{\pm0.2}$ | $39.5_{\pm0.3}$ | 44.9 |
| | FT | $\underline{79.2}_{\pm0.1}$ | $\underline{67.2}_{\pm0.2}$ | $59.8_{\pm0.2}$ | $\underline{20.2}_{\pm0.3}$ | $\underline{42.2}_{\pm0.2}$ | $\underline{47.4}$ |
| | Linear-probed | 74.7 | 62.8 | 55.3 | 24.8 | 37.0 | 45.0 |
| | Zero-shot | 63.2 | 56.4 | 68.4 | 32.4 | 41.2 | 49.6 |
| B/16 | Scratch-FT | $\underline{83.1}_{\pm0.0}$ | $\underline{73.1}_{\pm0.2}$ | $64.2_{\pm0.3}$ | $37.8_{\pm0.4}$ | $47.4_{\pm0.3}$ | 55.6 |
| | LP-FT | $82.1_{\pm0.1}$ | $70.8_{\pm0.2}$ | $62.2_{\pm0.2}$ | $35.4_{\pm0.4}$ | $44.6_{\pm0.2}$ | 53.3 |
| | FT | $82.8_{\pm0.1}$ | $72.6_{\pm0.3}$ | $\underline{68.5}_{\pm0.3}$ | $\underline{39.2}_{\pm0.3}$ | $\underline{48.0}_{\pm0.2}$ | $\underline{57.1}$ |
| | Linear-probed | 78.6 | 67.6 | 63.2 | 44.3 | 42.3 | 54.4 |
| | Zero-shot | 68.2 | 62.1 | 76.3 | 52.1 | 46.7 | 59.3 |
| L/14 | Scratch-FT | $86.2_{\pm0.1}$ | $\underline{77.0}_{\pm0.3}$ | $76.9_{\pm0.6}$ | $\underline{60.8}_{\pm0.7}$ | $\underline{57.9}_{\pm0.3}$ | 68.2 |
| | LP-FT | $84.5_{\pm0.0}$ | $73.9_{\pm0.1}$ | $73.8_{\pm0.2}$ | $55.8_{\pm0.5}$ | $54.5_{\pm0.2}$ | 64.5 |
| | FT | $85.4_{\pm0.1}$ | $75.5_{\pm0.1}$ | $\underline{80.4}_{\pm0.3}$ | $60.4_{\pm0.1}$ | $\underline{57.9}_{\pm0.1}$ | $\underline{68.6}$ |
| | Linear-probed | 83.0 | 73.1 | 77.4 | 65.4 | 53.5 | 67.4 |
| | Zero-shot | 75.2 | 69.9 | 86.9 | 71.8 | 58.6 | 71.8 |

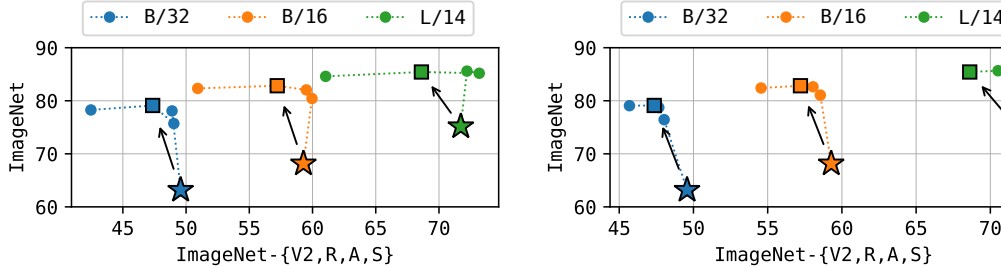

**Figure 7: Trade-off plots on ImageNet.** The star-shaped markers correspond to the zero-shot model, while the square-shaped markers denote the model fine-tuned with `50000` training steps and a learning rate of `1e-05`. It supplements Figure 2. **Left:** The number of training steps is kept at `50000`, while the learning rate varies as `1e-06`, `3e-06`, `1e-05`, and `3e-05` (the leftmost point denotes `3e-05`). **Right:** The learning rate is constant at `1e-05`, while the number of training steps is altered to `10000`, `30000`, `50000`, and `100000` (the leftmost point denotes `100000`).

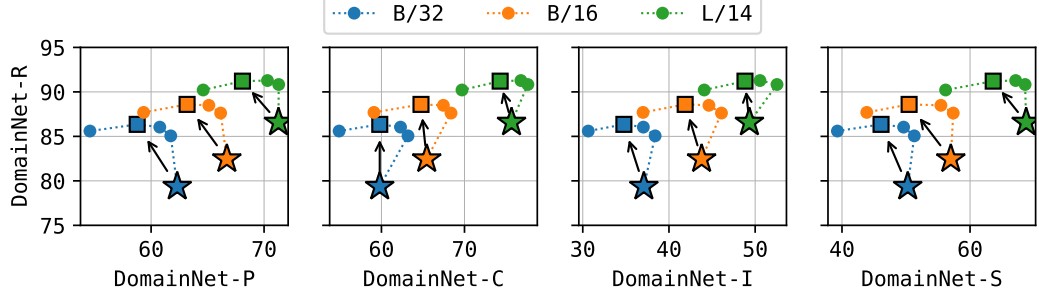

(a) DomainNet results with varying learning rates of `1e-06`, `3e-06`, `1e-05`, and `3e-05`.

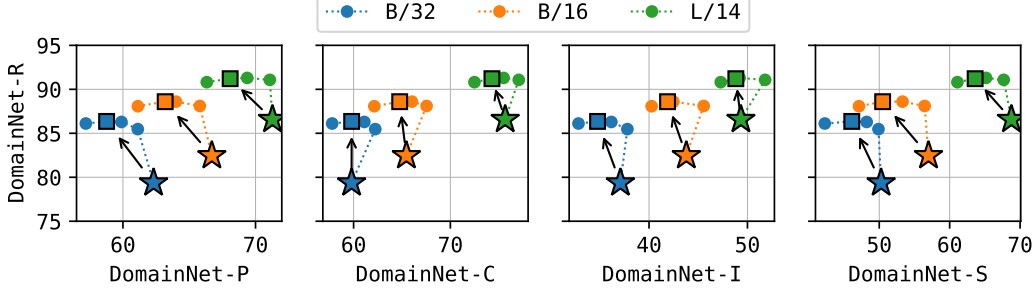

(b) DomainNet results with varying training steps of `1000`, `3000`, `5000`, and `10000`.

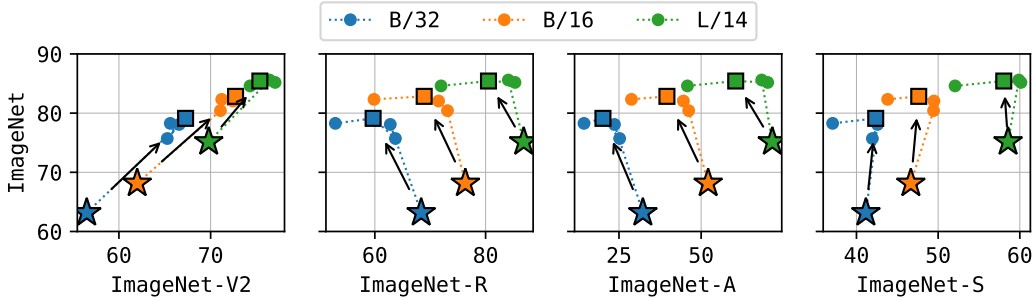

(c) ImageNet results with varying learning rates of `1e-06`, `3e-06`, `1e-05`, and `3e-05`.

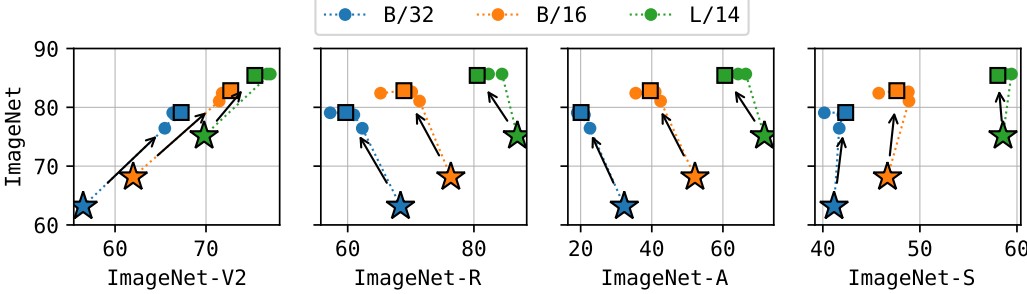

(d) ImageNet results with varying training steps of `10000`, `30000`, `50000`, and `100000`.

**Figure 8: Trade-off plots in detail.** This figure presents individual results for each distribution shift. See Figures 2 and 7 for results summarized in a single plot.

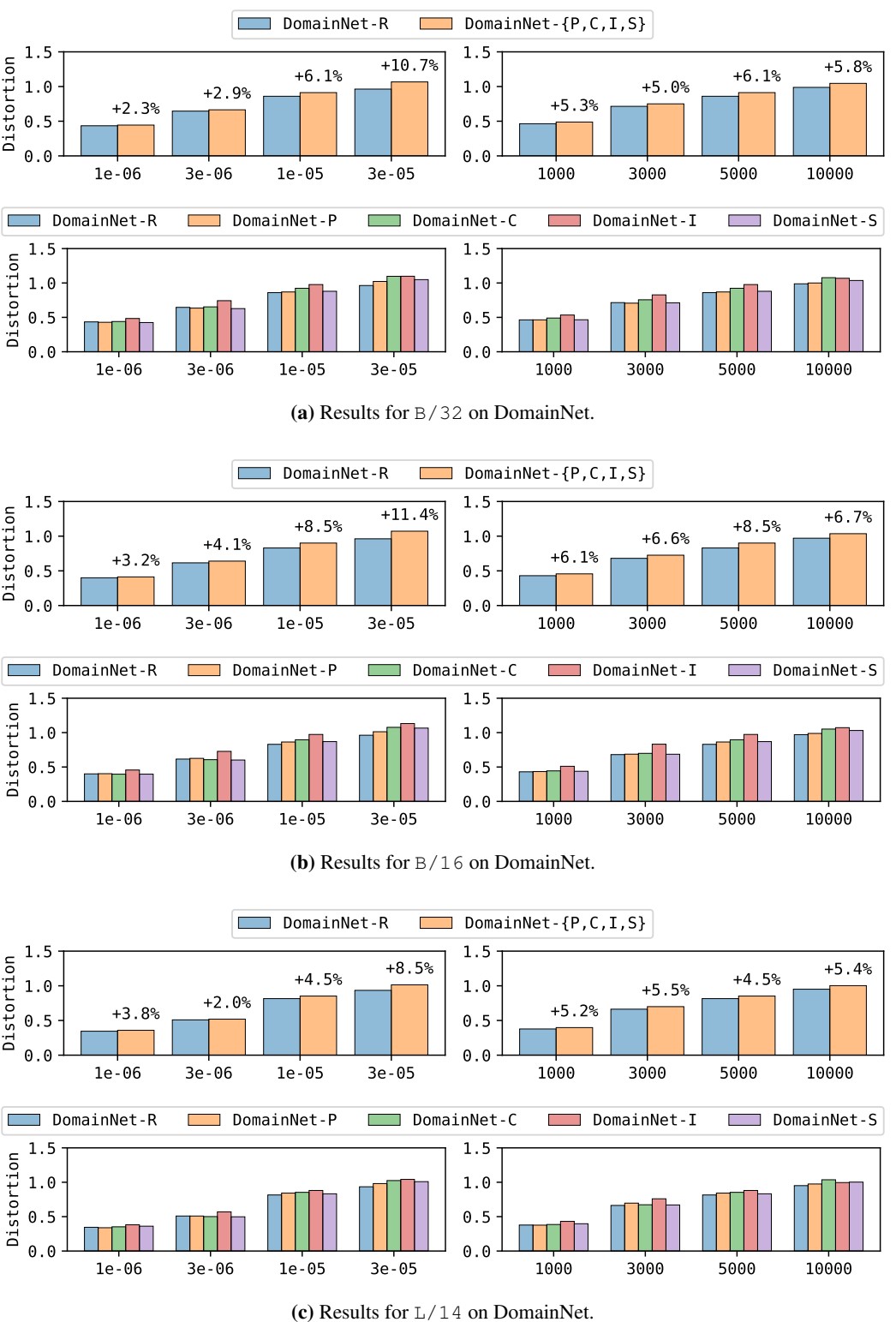

(a) Results for B/32 on DomainNet.

(b) Results for B/16 on DomainNet.

(c) Results for L/14 on DomainNet.

Figure 9: **Feature distortion on DomainNet in detail.** It supplements Figure 3 presented in the main text. The bottom plot for each subfigure presents individual results for each distribution shift.

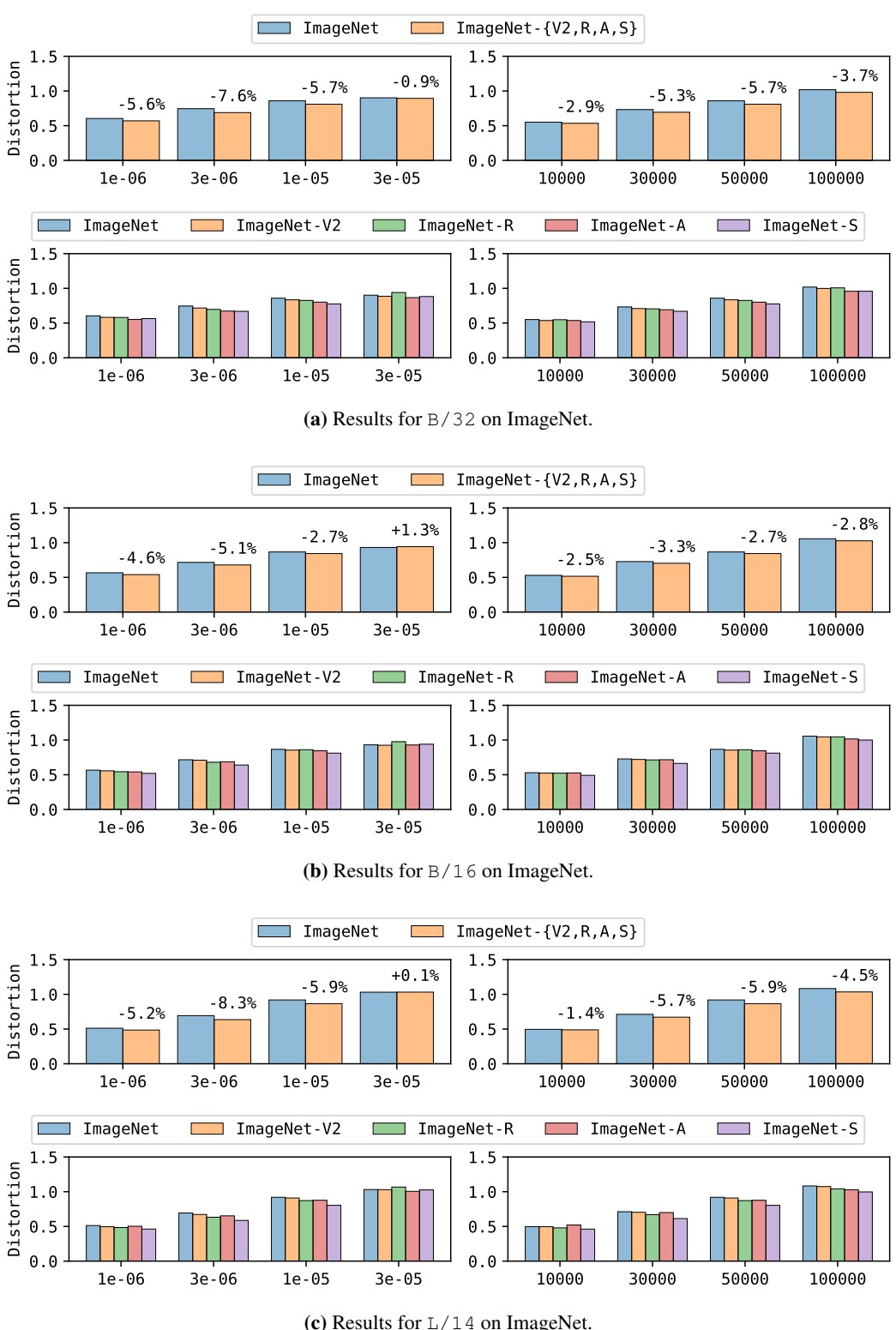

(a) Results for `B/32` on ImageNet.

(b) Results for `B/16` on ImageNet.

(c) Results for `L/14` on ImageNet.

Figure 10: **Feature distortion on ImageNet in detail.** It supplements Figure 3 presented in the main text. The bottom plot for each subfigure presents individual results for each distribution shift.

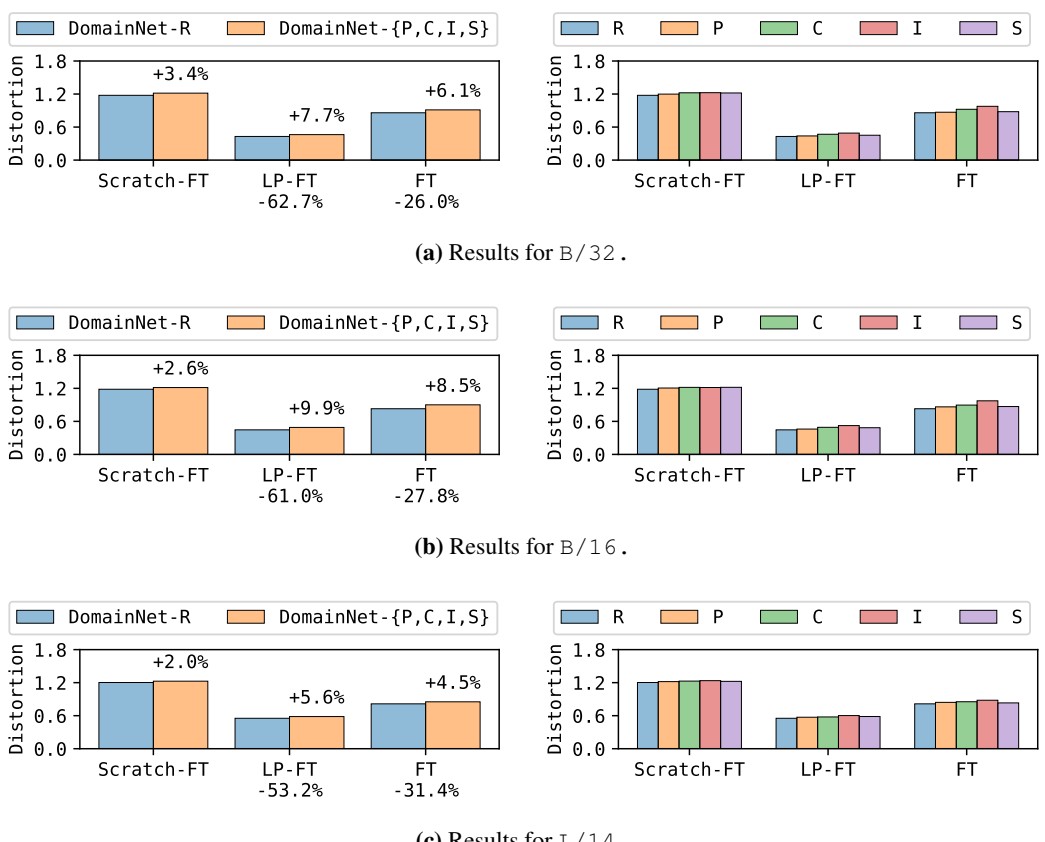

**Figure 11: LP-FT within the context of feature distortion theory on DomainNet.** The number on the bar denotes the relative difference in distortion values between reference and distribution shift data, while the value under the x-labels indicates the relative difference in distortion from the `Scratch-FT` method. **Left:** Results summarized in a single plot. **Right:** A detailed results for each dsitribution shift.

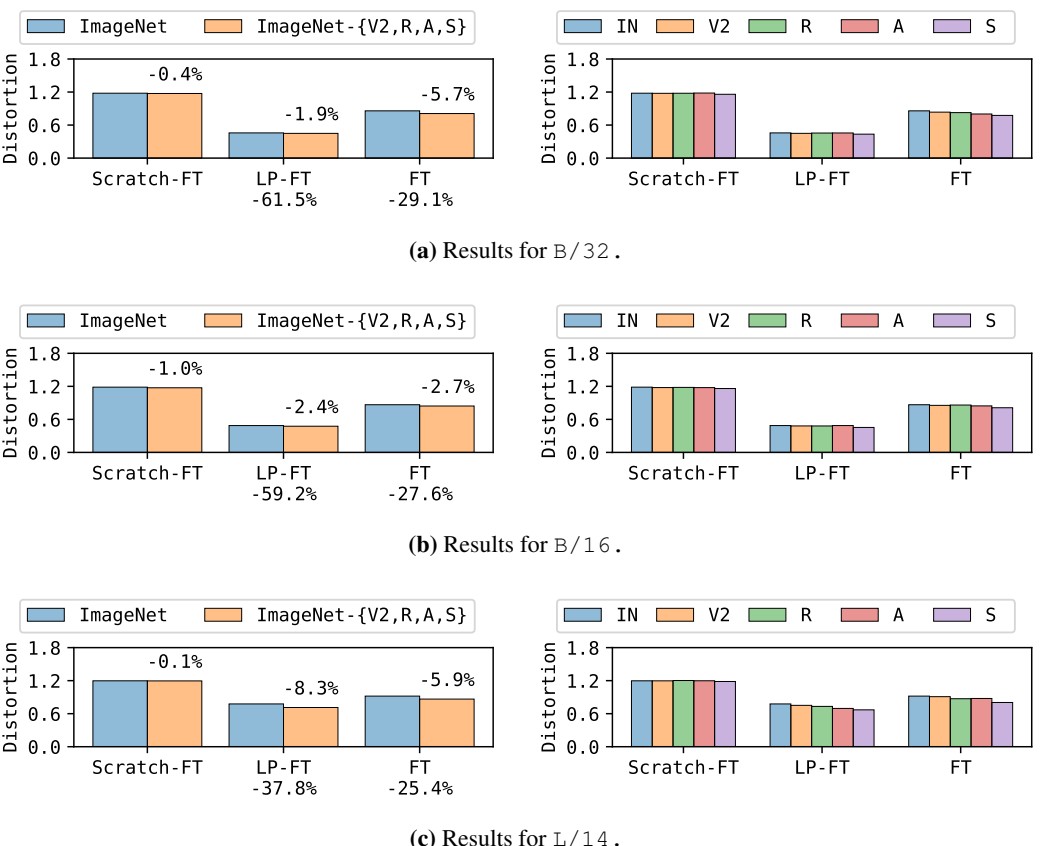

**(a)** Results for `B/32`.

**(b)** Results for `B/16`.

**(c)** Results for `L/14`.

**Figure 12: LP-FT within the context of feature distortion theory on ImageNet.** The number on the bar denotes the relative difference in distortion values between reference and distribution shift data, while the value under the x-labels indicates the relative difference in distortion from the `Scratch-FT` method. **Left:** Results summarized in a single plot. **Right:** A detailed results for each dsitribution shift.

### B.2 ROBUST FINE-TUNING

**Details on robust fine-tuning methods.** This paragraph provides a concise overview of the robust fine-tuning techniques we have considered in this work. For more comprehensive details, kindly consult the corresponding references:

- (L2 regularization towards Starting Point; Xuhong et al., 2018) L2SP applies an L2 regularization towards $\mathbf{\Theta}_0$, i.e., the update rule becomes

$$\mathbf{\Theta}_t = \mathbf{\Theta}_{t-1} - \eta \cdot \nabla_{\mathbf{\Theta}}(\hat{\mathcal{L}}_{\text{CE}}(\mathbf{\Theta}) + \lambda_{\text{L2SP}} \cdot \hat{\mathcal{R}}_{\text{L2SP}}(\mathbf{\Theta}))|_{\mathbf{\Theta}=\mathbf{\Theta}_{t-1}}, \tag{10}$$

where the regularization term is given by

$$\hat{\mathcal{R}}_{\text{L2SP}}(\mathbf{\Theta}) = \frac{1}{2} \cdot \|\mathbf{\Theta} - \mathbf{\Theta}_0\|_2^2. \tag{11}$$

We swept over $\lambda_{\text{L2SP}} \in \{\texttt{3e-03,1e-03,3e-04,1e-04}\}$ and set $\lambda_{\text{L2SP}} = \texttt{3e-04}$ both for DomainNet and ImageNet scenarios.

- (Knowledge Distillation; Hinton et al., 2015) KD applies a regularization towards $\mathbf{\Theta}_0$ using the update rule

$$\mathbf{\Theta}_t = \mathbf{\Theta}_{t-1} - \eta \cdot \nabla_{\mathbf{\Theta}}(\hat{\mathcal{L}}_{\text{CE}}(\mathbf{\Theta}) + \lambda_{\text{KD}} \cdot \hat{\mathcal{R}}_{\text{KD}}(\mathbf{\Theta}))|_{\mathbf{\Theta}=\mathbf{\Theta}_{t-1}}, \tag{12}$$

where the regularization term is given by the distillation loss,

$$\hat{\mathcal{R}}_{\text{KD}}(\mathbf{\Theta}) = -\sum_{k=1}^{K} \text{Softmax}^{(k)}(\mathbf{W}_0 \mathcal{F}_{\boldsymbol{\theta}_0}(\boldsymbol{x})) \cdot \log \text{Softmax}^{(k)}(\mathbf{W}\mathcal{F}_{\boldsymbol{\theta}}(\boldsymbol{x})). \tag{13}$$

We swept over $\lambda_{\text{KD}} \in \{\texttt{0.1,0.2,0.5,1.0,2.0,5.0}\}$ and set $\lambda_{\text{KD}} = \texttt{0.1}$ both for DomainNet and ImageNet scenarios.

- (Context-Aware Robust Fine-Tuning; Mao et al., 2023) CAR-FT applies a regularization towards $\mathbf{\Theta}_0$ using the update rule

$$\mathbf{\Theta}_t = \mathbf{\Theta}_{t-1} - \eta \cdot \nabla_{\mathbf{\Theta}}(\hat{\mathcal{L}}_{\text{CE}}(\mathbf{\Theta}) + \lambda_{\text{CAR-FT}} \cdot \hat{\mathcal{R}}_{\text{CAR-FT}}(\boldsymbol{\theta}))|_{\mathbf{\Theta}=\mathbf{\Theta}_{t-1}}, \tag{14}$$

where the regularization term is defined for the vision model parameters $\boldsymbol{\theta}$,

$$\hat{\mathcal{R}}_{\text{CAR-FT}}(\boldsymbol{\theta}) = -\sum_{m=1}^{M} \text{Softmax}^{(m)}(\mathbf{W}_{\text{CTX}} \mathcal{F}_{\boldsymbol{\theta}_0}(\boldsymbol{x})) \cdot \log \text{Softmax}^{(m)}(\mathbf{W}_{\text{CTX}} \mathcal{F}_{\boldsymbol{\theta}}(\boldsymbol{x})). \tag{15}$$

We swept over $\lambda_{\text{CAR-FT}} \in \{\texttt{0.1,0.2,0.5,1.0,2.0,5.0}\}$ and set $\lambda_{\text{CAR-FT}} = 1.0$ both for DomainNet and ImageNet scenarios.

- (Exponential Moving Average) EMA takes an exponential moving average over the optimization trajectory, i.e., for every $t^{\text{th}}$ iteration,

$$\boldsymbol{\theta}_{\text{EMA}} \leftarrow \lambda_{\text{EMA}} \cdot \boldsymbol{\theta}_{\text{EMA}} + (1 - \lambda_{\text{EMA}}) \cdot \boldsymbol{\theta}_t, \tag{16}$$

using the decay factor $\lambda_{\text{EMA}} \in [0, 1]$, where we initialized the buffer $\boldsymbol{\theta}_{\text{EMA}} \leftarrow \boldsymbol{\theta}_0$. Instead of the fine-tuned solution $\boldsymbol{\theta}_T$, EMA utilizes $\theta_{\text{EMA}}$ for evaluation purposes. We swept over $\lambda_{\text{EMA}} \in \{\texttt{0.9999,0.9997,0.9995}\}$ and set $\lambda_{\text{EMA}} = \texttt{0.9995}$ for DomainNet and $\lambda_{\text{EMA}} = \texttt{0.9999}$ for ImageNet.

- (Weight-Space Ensembles for Fine-Tuning; Wortsman et al., 2022b) WiSE is a post-hoc simple weight averaging scheme that takes an weighted average between the zero-shot parameters $\boldsymbol{\theta}_0$ and the fine-tuned parameters $\boldsymbol{\theta}_T$,

$$\boldsymbol{\theta}_{\text{WiSE}} = (1 - \lambda_{\text{WiSE}}) \cdot \boldsymbol{\theta}_0 + \lambda_{\text{WiSE}} \cdot \boldsymbol{\theta}_T, \tag{17}$$

using the mixing coefficient $\lambda_{\text{WiSE}} \in [0, 1]$. Unless otherwise mentioned in the main text, we used $\lambda_{\text{WiSE}} = 0.9$ because it performs satisfactorily on the reference data, as demonstrated in Tables 6 and 7.

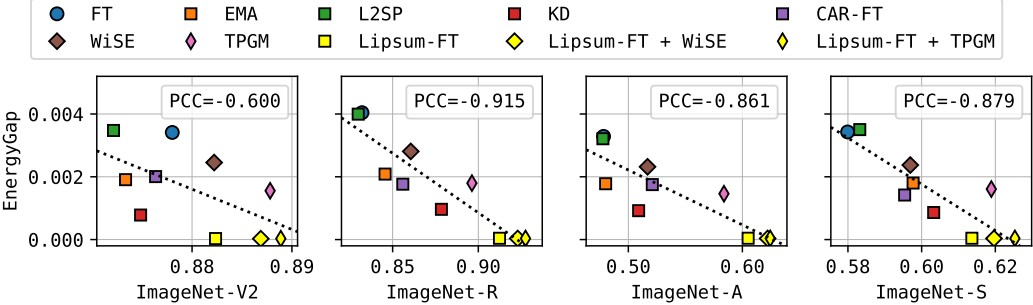

**Figure 13: Energy gaps in robust fine-tuning baselines on ImageNet.** It supplements Figure 4.

- (Trainable Projected Gradient Method; Tian et al., 2023) `TPGM` is a post-hoc optimization scheme that takes a layer-wise projection of the fine-tuned parameters to be within the constraint $\|\boldsymbol{\theta}_T - \boldsymbol{\theta}_0\|_2 \leq \gamma_{\text{TPGM}}$ in the layer-wise manner. Let $\boldsymbol{\theta}_T$ be the fine-tuned parameters and $\boldsymbol{\theta}_0$ be the zero-shot parameters for each layer, with a slight abuse of notation. Then, `TPGM` defines the projected parameters as

$$\boldsymbol{\theta}_{\text{TPGM}} = \boldsymbol{\theta}_0 + \frac{1}{\max\left(1, \|\boldsymbol{\theta}_T - \boldsymbol{\theta}_0\|_2 / \gamma_{\text{TPGM}}\right)} \cdot (\boldsymbol{\theta}_T - \boldsymbol{\theta}_0), \tag{18}$$

and carries out the additional optimization using the validation data $\mathcal{D}_{\text{Val}}$,

$$\gamma_{\text{TPGM}}^* = \arg\min_{\gamma_{\text{TPGM}}} \sum_{(\boldsymbol{x},k) \in \mathcal{D}_{\text{Val}}} -\log \text{Softmax}^{(k)}(\mathbf{W}\mathcal{F}_{\boldsymbol{\theta}_{\text{TPGM}}}). \tag{19}$$

Following the implementation detailed by Tian et al. (2023), we carried out this optimization process using the Adam optimizer. The learning rate was set to `1e-02`, and the number of training steps was `200`. Additionally, we experimented with `TPGM-C`, a controlled variant of `TPGM`, where regularization over $\gamma_{\text{TPGM}}$ was introduced during the optimization.

Caveat: To streamline the experiments, we employed the test split of `DomainNet-R` as $\mathcal{D}_{\text{Val}}$ in the DomainNet scenario. Similarly, in the ImageNet setup, we used the validation split of `ImageNet` as $\mathcal{D}_{\text{Val}}$. Thus, we note here that it is challenging to categorize the reference data performance of the `TPGM` presented in the main text purely as *test* performance, raising a point of contention. However, we would like to emphasize that there are no issues with the experimental comparison.

- (Model Soups; Wortsman et al., 2022a) `Soup` and `Ensemble` refer to the greedy soup and greedy ensemble algorithms introduced in Wortsman et al. (2022a). These techniques operate in a scenario where multiple models are created with varying hyperparameter setups. In contrast to the conventional approach of selecting the best-performing model from this set, `Soup` and `Ensemble` respectively attain improved performance by averaging their weights and ensembling their outputs.

  In our experiment, we varied the following three hyperparameters; (1) mini-batch sizes $\{128, 160, 192, 224, 256\}$, (2) initial learning rates $\{1e-06, 2e-06, \ldots, 3e-05\}$, and (3) the number of training iterations $\{3000, 5000, 7000\}$ for the DomainNet scenario and $\{30000, 50000, 70000\}$ for the ImageNet scenario.

  Caveat: Similar to the situation with `TPGM`, it is hard to label the reference data performance of the `Soup` and `Ensemble` methods discussed in the main text solely as *test* performance since the test data have been involved in the creation process of those methods. Also, it is worth mentioning that we have conducted additional hyperparameter searches as well. Again, however, we want to highlight that there are no concerns regarding the experimental comparison.

**Supplementary plots for Figure 4.** In Figure 13, we present the same plot for the ImageNet scenario. As discussed in § 4.2, it clearly depicts a distinct correlation between the energy gap and the robustness to distribution shifts.

**Table 5: Full results on distribution shift tasks.** It summarizes the accuracy of various methods for (a) `B/32` and (b) `L/14` models in distribution shift scenarios on DomainNet and ImageNet (for `B/16`, please refer to Table 1 presented in the main text). All fine-tuning results are averaged from five measurements, and the corresponding standard deviation values are provided. An underline highlights the top two values in each column.

**(a)** `B/32`.

| Method | DomainNet | | | | | ImageNet | | | | |
|---|---|---|---|---|---|---|---|---|---|---|
| | R | P | C | I | S | IN | V2 | R | A | S |
| Zero-shot | 79.4 | 62.4 | 59.8 | 37.1 | 50.3 | 63.2 | 56.4 | 68.4 | 32.4 | 41.2 |
| FT | 86.5±0.1 | 58.9±0.2 | 59.7±0.3 | 34.8±0.2 | 45.9±0.3 | 79.2±0.1 | 67.2±0.2 | 59.8±0.2 | 20.2±0.3 | 42.2±0.2 |
| EMA | 86.8±0.1 | 61.0±0.2 | 62.1±0.2 | 37.3±0.2 | 48.9±0.3 | 79.1±0.1 | 67.4±0.2 | 60.8±0.4 | 19.9±0.3 | 42.7±0.4 |
| L2SP | 86.4±0.1 | 58.9±0.2 | 59.8±0.1 | 35.0±0.3 | 46.0±0.2 | 79.2±0.1 | 67.3±0.1 | 60.1±0.3 | 20.6±0.3 | 42.3±0.3 |
| KD | 86.6±0.1 | 60.0±0.2 | 60.9±0.4 | 36.0±0.3 | 47.3±0.3 | 79.4±0.1 | 68.0±0.1 | 63.7±0.2 | 22.2±0.2 | 43.5±0.3 |
| CAR-FT | 86.7±0.1 | 60.0±0.2 | 60.4±0.4 | 36.1±0.2 | 47.1±0.3 | 79.5±0.0 | 67.8±0.2 | 62.4±0.2 | 23.2±0.2 | 43.6±0.2 |
| Lipsum-FT | 86.6±0.1 | 61.9±0.1 | 62.1±0.1 | 38.4±0.1 | 49.6±0.2 | 79.4±0.1 | 68.4±0.1 | 67.2±0.2 | 27.7±0.3 | 45.0±0.3 |

**(b)** `L/14`.

| Method | DomainNet | | | | | ImageNet | | | | |
|---|---|---|---|---|---|---|---|---|---|---|
| | R | P | C | I | S | IN | V2 | R | A | S |
| Zero-shot | 86.6 | 71.3 | 75.7 | 49.3 | 68.8 | 75.2 | 69.9 | 86.9 | 71.8 | 58.6 |
| FT | 91.2±0.1 | 68.1±0.2 | 74.3±0.4 | 48.6±0.6 | 63.5±0.5 | 85.4±0.1 | 75.5±0.1 | 80.4±0.3 | 60.4±0.1 | 57.9±0.1 |
| EMA | 91.5±0.0 | 70.8±0.5 | 76.8±0.3 | 51.2±0.6 | 67.3±0.7 | 86.2±0.1 | 77.1±0.3 | 83.6±0.7 | 64.9±0.7 | 60.1±0.3 |
| L2SP | 91.2±0.0 | 68.9±0.2 | 74.8±0.2 | 48.9±0.3 | 63.9±0.4 | 85.8±0.1 | 76.3±0.2 | 81.9±0.3 | 62.7±0.5 | 58.5±0.2 |
| KD | 91.3±0.1 | 70.0±0.2 | 75.8±0.2 | 49.5±0.5 | 65.9±0.4 | 86.0±0.1 | 77.0±0.1 | 85.2±0.2 | 65.5±0.4 | 60.0±0.2 |
| CAR-FT | 91.4±0.1 | 69.8±0.1 | 76.0±0.3 | 50.2±0.5 | 66.1±0.3 | 86.3±0.1 | 76.8±0.1 | 84.2±0.3 | 66.6±0.2 | 60.0±0.1 |
| Lipsum-FT | 91.3±0.0 | 71.3±0.1 | 77.2±0.1 | 51.9±0.2 | 68.2±0.2 | 86.1±0.0 | 77.7±0.2 | 87.0±0.1 | 71.7±0.3 | 61.2±0.2 |

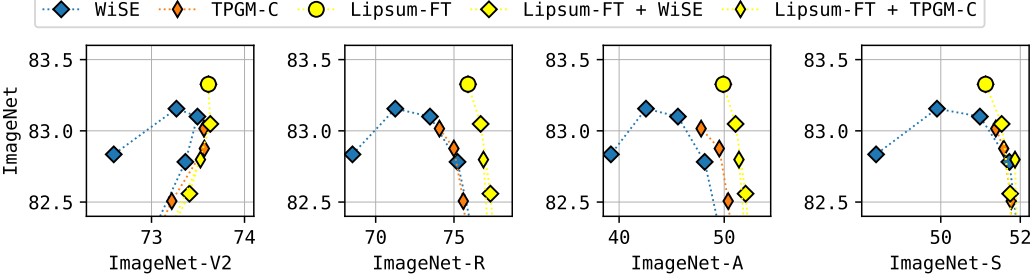

**Figure 14: Scatter plots for post-hoc methods on ImageNet.** It supplements Figure 5. See Table 7 for numerical results with standard deviations.

## B.3 Lipsum-FT

**Full table of results for Table 1.** In Table 5, we performed additional comparative assessments of robust fine-tuning techniques using both `B/32` and `L/14` architectures. For `L/14`, experiments were conducted using a batch size of `128` due to hardware memory constraints. We also applied the same hyperparameters from `B/16` to `B/32` and `L/14` without any adjustments. Overall, our proposed `Lipsum-FT` approach demonstrates enhanced distribution shift performance while maintaining performance in the reference data.

**Supplementary plots and tables for Figure 5.** In Figure 14, we present the same plots for the ImageNet scenario. Tables 6 and 7 further provides the numerical results averaged over five measurements along with its standard deviation values. Again, `Lipsum-FT` exhibits superior performance both on the reference and distribution shift data. We could further enhance the performance in distribution shifts by combining ours with the existing post-hoc methods `WiSE` and `TPGM`.

**Table 6: Results for post-hoc methods on DomainNet.** It supplements Figure 5. An underline highlights the top two values in each column for each group.

| Method | R | P | C | I | S |
|---|---|---|---|---|---|
| WiSE ($\lambda_{\texttt{WiSE}}$ = 1.0) | 88.6±0.1 | 63.1±0.2 | 65.0±0.4 | 41.9±0.3 | 51.2±0.5 |
| WiSE ($\lambda_{\texttt{WiSE}}$ = 0.9) | 88.8±0.1 | 64.5±0.1 | 66.5±0.4 | 43.6±0.3 | 53.2±0.4 |
| WiSE ($\lambda_{\texttt{WiSE}}$ = 0.8) | 88.8±0.0 | 65.7±0.1 | 67.6±0.3 | 44.9±0.2 | 54.9±0.4 |
| WiSE ($\lambda_{\texttt{WiSE}}$ = 0.7) | 88.6±0.0 | 66.7±0.2 | 68.4±0.2 | 46.0±0.2 | 56.3±0.3 |
| WiSE ($\lambda_{\texttt{WiSE}}$ = 0.6) | 88.3±0.1 | 67.4±0.2 | 68.9±0.2 | 46.6±0.2 | 57.2±0.3 |
| WiSE ($\lambda_{\texttt{WiSE}}$ = 0.5) | 87.8±0.1 | 67.9±0.2 | 69.0±0.2 | 47.0±0.1 | 57.9±0.2 |
| Lipsum-FT + WiSE ($\lambda_{\texttt{WiSE}}$ = 1.0) | 89.0±0.0 | 66.3±0.2 | 68.0±0.0 | 46.0±0.2 | 56.2±0.2 |
| Lipsum-FT + WiSE ($\lambda_{\texttt{WiSE}}$ = 0.9) | 88.8±0.0 | 66.8±0.2 | 68.4±0.1 | 46.4±0.1 | 56.9±0.2 |
| Lipsum-FT + WiSE ($\lambda_{\texttt{WiSE}}$ = 0.8) | 88.5±0.0 | 67.2±0.2 | 68.6±0.0 | 46.7±0.1 | 57.5±0.2 |
| Lipsum-FT + WiSE ($\lambda_{\texttt{WiSE}}$ = 0.7) | 88.1±0.1 | 67.5±0.1 | 68.7±0.1 | 46.9±0.1 | 57.8±0.2 |
| Lipsum-FT + WiSE ($\lambda_{\texttt{WiSE}}$ = 0.6) | 87.7±0.0 | 67.7±0.1 | 68.6±0.1 | 46.8±0.1 | 58.1±0.2 |
| Lipsum-FT + WiSE ($\lambda_{\texttt{WiSE}}$ = 0.5) | 87.1±0.0 | 67.8±0.1 | 68.4±0.1 | 46.7±0.1 | 58.3±0.2 |
| TPGM | 89.0±0.0 | 65.8±0.2 | 67.3±0.1 | 44.9±0.3 | 55.4±0.1 |
| TPGM-C (reg=0.1) | 88.9±0.0 | 65.8±0.2 | 67.3±0.2 | 44.9±0.3 | 55.4±0.2 |
| TPGM-C (reg=0.2) | 88.9±0.1 | 65.9±0.2 | 67.3±0.2 | 44.9±0.3 | 55.4±0.1 |
| TPGM-C (reg=0.5) | 88.9±0.0 | 66.0±0.2 | 67.4±0.2 | 45.1±0.2 | 55.5±0.1 |
| TPGM-C (reg=1.0) | 88.8±0.1 | 66.1±0.2 | 67.6±0.2 | 45.4±0.1 | 55.8±0.2 |
| TPGM-C (reg=2.0) | 88.7±0.0 | 66.6±0.1 | 68.1±0.3 | 46.2±0.1 | 56.4±0.1 |
| TPGM-C (reg=5.0) | 88.1±0.1 | 67.4±0.2 | 68.6±0.1 | 46.9±0.1 | 57.4±0.1 |
| Lipsum-FT + TPGM | 89.0±0.1 | 66.4±0.2 | 68.1±0.1 | 46.2±0.2 | 56.6±0.2 |
| Lipsum-FT + TPGM-C (reg=0.1) | 89.0±0.1 | 66.5±0.2 | 68.1±0.1 | 46.2±0.1 | 56.6±0.3 |
| Lipsum-FT + TPGM-C (reg=0.2) | 89.0±0.1 | 66.4±0.2 | 68.1±0.1 | 46.2±0.1 | 56.6±0.2 |
| Lipsum-FT + TPGM-C (reg=0.5) | 89.0±0.0 | 66.5±0.2 | 68.1±0.1 | 46.2±0.1 | 56.6±0.2 |
| Lipsum-FT + TPGM-C (reg=1.0) | 89.0±0.0 | 66.5±0.2 | 68.2±0.1 | 46.3±0.2 | 56.7±0.3 |
| Lipsum-FT + TPGM-C (reg=2.0) | 88.8±0.1 | 66.7±0.2 | 68.3±0.1 | 46.6±0.2 | 56.9±0.2 |
| Lipsum-FT + TPGM-C (reg=5.0) | 88.0±0.0 | 67.3±0.1 | 68.5±0.2 | 46.9±0.1 | 57.6±0.1 |

**Hyperparameters in `Lipsum-FT`.** There are two hyperparameters in `Lipsum-FT`: (1) a length of randomly generated text tokens $L$ and (2) the number of text tokens $M$ to be generated for each fine-tuning iteration. Table 8 provides ablation results regarding these hyperparameters, indicating that both a substantial increase and decrease in the length of the random text can result in a decline in performance, as illustrated in cases where $L = 1$ and $L = 32$. It implies that the effectiveness of text guidance can fluctuate depending on the input text and further motivates future works aimed at creating a novel input text for robust fine-tuning. We used $L = 8$ and $M = 80$ in the main text.

**Connection between KLD and MSE.** For `CAR-FT` and `Lipsum-FT`$_{\text{KLD}}$ methods presented in Tables 3 and 9, we utilized the following KLD loss formulation,

$$\hat{\mathcal{R}}_{\text{KLD}}^{\tau} = -\tau^2 \cdot \sum_{m=1}^{M} \text{Softmax}^{(m)}(\boldsymbol{v}_{\boldsymbol{\theta}_0,\boldsymbol{\phi}}/\tau) \cdot \text{LogSoftmax}^{(m)}(\boldsymbol{v}_{\boldsymbol{\theta},\boldsymbol{\phi}}/\tau), \qquad (20)$$

where $\tau$ is a temperature hyperparameter for smoothing categorical outputs (Hinton et al., 2015). Although we kept this temperature fixed at $\tau = 1$ throughout all experiments, it serves as a bridge connecting the KLD loss and the MSE loss. More precisely, taking a limit $\tau \to \infty$ to the KLD loss $\hat{\mathcal{R}}_{\text{KLD}}^{\tau}$ gives us the following (Kim et al., 2021):

$$\lim_{\tau \to \infty} \hat{\mathcal{R}}_{\text{KLD}}^{\tau} = \frac{1}{2M} \|\boldsymbol{v}_{\boldsymbol{\theta},\boldsymbol{\phi}} - \boldsymbol{v}_{\boldsymbol{\theta}_0,\boldsymbol{\phi}}\|_2^2 - \frac{1}{2M^2}\left(\sum_{i=1}^{M} \boldsymbol{v}_{\boldsymbol{\theta},\boldsymbol{\phi}}^{(i)} - \sum_{j=1}^{M} \boldsymbol{v}_{\boldsymbol{\theta}_0,\boldsymbol{\phi}}^{(j)}\right) + \text{Constant}. \qquad (21)$$

The first term aligns with our regularization defined in Equation 7, which can be understood as logit matching (Ba & Caruana, 2014). Here, the second term causes an increase in the values in $\boldsymbol{v}_{\boldsymbol{\theta},\boldsymbol{\phi}}$ and hinders complete logit matching as pointed out in Kim et al. (2021). In the context of our `Lipsum-FT` framework, the scale of $\boldsymbol{v}_{\boldsymbol{\theta},\boldsymbol{\phi}}$ contains the energy information associated with the pre-trained relationship between vision and language models. Consequently, MSE is a better choice than KLD from this perspective, as validated by empirical results in Tables 3 and 9; `CAR-FT`$_{\text{MSE}}$ surpasses `CAR-FT`, and `Lipsum-FT` outperforms `Lipsum-FT`$_{\text{KLD}}$.

**Table 7: Results for post-hoc methods on ImageNet.** It supplements Figure 14. An underline highlights the top two values in each column for each group.

| Method | IN | V2 | R | A | S |
|---|---|---|---|---|---|
| WiSE ($\lambda_{\texttt{WiSE}}$ = 1.0) | 82.8±0.1 | 72.6±0.3 | 68.5±0.3 | 39.2±0.3 | 48.0±0.2 |
| WiSE ($\lambda_{\texttt{WiSE}}$ = 0.9) | 83.2±0.0 | 73.3±0.2 | 71.3±0.2 | 42.6±0.3 | 49.9±0.2 |
| WiSE ($\lambda_{\texttt{WiSE}}$ = 0.8) | 83.1±0.1 | 73.5±0.2 | 73.5±0.2 | 45.6±0.4 | 51.2±0.1 |
| WiSE ($\lambda_{\texttt{WiSE}}$ = 0.7) | 82.8±0.1 | 73.4±0.2 | 75.2±0.2 | 48.1±0.5 | 52.2±0.1 |
| WiSE ($\lambda_{\texttt{WiSE}}$ = 0.6) | 82.1±0.1 | 72.9±0.1 | 76.6±0.2 | 50.3±0.3 | 52.5±0.1 |
| WiSE ($\lambda_{\texttt{WiSE}}$ = 0.5) | 80.9±0.0 | 72.1±0.1 | 77.6±0.1 | 51.8±0.4 | 52.5±0.1 |
| Lipsum-FT + WiSE ($\lambda_{\texttt{WiSE}}$ = 1.0) | 83.3±0.0 | 73.6±0.1 | 75.9±0.1 | 49.9±0.3 | 51.4±0.1 |
| Lipsum-FT + WiSE ($\lambda_{\texttt{WiSE}}$ = 0.9) | 83.0±0.0 | 73.6±0.0 | 76.7±0.1 | 51.1±0.4 | 51.9±0.1 |
| Lipsum-FT + WiSE ($\lambda_{\texttt{WiSE}}$ = 0.8) | 82.6±0.0 | 73.4±0.1 | 77.3±0.1 | 52.0±0.2 | 52.2±0.1 |
| Lipsum-FT + WiSE ($\lambda_{\texttt{WiSE}}$ = 0.7) | 81.8±0.1 | 72.9±0.1 | 77.8±0.1 | 52.8±0.1 | 52.2±0.1 |
| Lipsum-FT + WiSE ($\lambda_{\texttt{WiSE}}$ = 0.6) | 80.9±0.1 | 72.1±0.2 | 78.1±0.1 | 53.4±0.2 | 52.1±0.1 |
| Lipsum-FT + WiSE ($\lambda_{\texttt{WiSE}}$ = 0.5) | 79.6±0.1 | 71.1±0.1 | 78.2±0.1 | 53.7±0.2 | 51.8±0.1 |
| TPGM | 83.0±0.1 | 73.6±0.1 | 74.1±0.2 | 47.8±0.6 | 51.7±0.2 |
| TPGM-C (reg=0.1) | 82.9±0.1 | 73.6±0.1 | 75.0±0.2 | 49.5±0.5 | 52.0±0.1 |
| TPGM-C (reg=0.2) | 82.5±0.1 | 73.2±0.1 | 75.6±0.2 | 50.4±0.4 | 52.2±0.2 |
| TPGM-C (reg=0.5) | 81.3±0.1 | 72.3±0.1 | 76.7±0.2 | 52.0±0.1 | 52.1±0.1 |
| TPGM-C (reg=1.0) | 78.8±0.1 | 70.4±0.1 | 77.6±0.1 | 53.0±0.3 | 51.4±0.1 |
| TPGM-C (reg=2.0) | 75.2±0.1 | 67.7±0.2 | 77.9±0.1 | 53.2±0.2 | 50.1±0.1 |
| TPGM-C (reg=5.0) | 71.3±0.0 | 64.6±0.1 | 77.3±0.1 | 52.8±0.1 | 48.3±0.0 |
| Lipsum-FT + TPGM | 82.8±0.0 | 73.5±0.1 | 76.9±0.1 | 51.4±0.3 | 52.3±0.1 |
| Lipsum-FT + TPGM-C (reg=0.1) | 82.2±0.1 | 73.1±0.1 | 77.2±0.1 | 52.0±0.2 | 52.4±0.1 |
| Lipsum-FT + TPGM-C (reg=0.2) | 81.6±0.1 | 72.7±0.1 | 77.4±0.1 | 52.5±0.3 | 52.4±0.1 |
| Lipsum-FT + TPGM-C (reg=0.5) | 79.8±0.1 | 71.3±0.3 | 77.6±0.1 | 53.0±0.1 | 52.0±0.1 |
| Lipsum-FT + TPGM-C (reg=1.0) | 77.2±0.0 | 69.3±0.2 | 77.5±0.1 | 53.1±0.1 | 51.0±0.1 |
| Lipsum-FT + TPGM-C (reg=2.0) | 73.7±0.0 | 66.6±0.1 | 77.1±0.1 | 52.9±0.2 | 49.3±0.1 |
| Lipsum-FT + TPGM-C (reg=5.0) | 70.6±0.0 | 64.1±0.0 | 76.7±0.0 | 52.5±0.1 | 47.8±0.1 |

**Table 8: Ablation results on hyperparameters.** There are two hyperparameters $L$ and $M$ in Lipsum-FT. An underline highlights the top two values in each column for each group.

| $L$ | $M$ | DomainNet | | | | | ImageNet | | | | |
|---|---|---|---|---|---|---|---|---|---|---|---|
| | | R | P | C | I | S | IN | V2 | R | A | S |
| 1 | 80 | 89.0±0.0 | 66.0±0.1 | 67.7±0.1 | 45.4±0.1 | 55.6±0.1 | 83.4±0.1 | 73.7±0.1 | 74.9±0.1 | 48.4±0.4 | 51.1±0.1 |
| 2 | 80 | 89.0±0.0 | 66.1±0.1 | 67.7±0.3 | 45.9±0.1 | 55.9±0.2 | 83.4±0.0 | 73.7±0.1 | 75.5±0.1 | 49.0±0.3 | 51.4±0.1 |
| 4 | 80 | 89.0±0.1 | 66.3±0.1 | 68.0±0.1 | 46.0±0.1 | 56.1±0.1 | 83.3±0.0 | 73.6±0.1 | 75.9±0.1 | 49.5±0.2 | 51.4±0.2 |
| 8 | 80 | 89.0±0.0 | 66.3±0.2 | 68.0±0.2 | 46.0±0.2 | 56.2±0.1 | 83.2±0.1 | 73.5±0.2 | 75.8±0.1 | 49.5±0.1 | 51.4±0.1 |
| 16 | 80 | 89.0±0.1 | 66.1±0.2 | 67.7±0.2 | 45.7±0.2 | 55.6±0.1 | 83.4±0.1 | 73.8±0.2 | 75.1±0.1 | 48.6±0.5 | 51.3±0.2 |
| 32 | 80 | 88.9±0.0 | 66.1±0.2 | 67.7±0.0 | 45.7±0.2 | 55.9±0.2 | 83.4±0.0 | 73.6±0.1 | 74.6±0.1 | 48.6±0.4 | 51.0±0.1 |
| 8 | 10 | 88.9±0.1 | 66.4±0.2 | 67.9±0.1 | 46.3±0.2 | 56.3±0.4 | 83.2±0.1 | 73.5±0.2 | 75.8±0.1 | 49.5±0.1 | 51.4±0.1 |
| 8 | 20 | 88.9±0.0 | 66.4±0.1 | 68.1±0.1 | 46.3±0.2 | 56.4±0.3 | 83.2±0.0 | 73.4±0.2 | 75.8±0.1 | 49.7±0.6 | 51.5±0.1 |
| 8 | 40 | 88.9±0.0 | 66.3±0.1 | 68.0±0.2 | 46.1±0.2 | 56.4±0.3 | 83.3±0.1 | 73.6±0.1 | 75.8±0.1 | 50.0±0.3 | 51.4±0.1 |
| 8 | 80 | 89.0±0.0 | 66.3±0.2 | 68.0±0.0 | 46.0±0.2 | 56.2±0.2 | 83.3±0.0 | 73.6±0.1 | 75.9±0.1 | 49.9±0.3 | 51.4±0.1 |
| 8 | 160 | 89.0±0.0 | 66.3±0.1 | 68.0±0.2 | 46.2±0.1 | 56.3±0.2 | 83.3±0.0 | 73.7±0.2 | 75.9±0.1 | 49.8±0.4 | 51.4±0.1 |
| 8 | 320 | 89.0±0.1 | 66.3±0.1 | 68.0±0.2 | 46.2±0.2 | 56.3±0.2 | 83.3±0.1 | 73.6±0.1 | 76.0±0.2 | 50.0±0.2 | 51.4±0.2 |

**Allowing certain level of feature distortion.** One can notice that a distinction might exist between features before and after fine-tuning, i.e., $\mathcal{F}_{\theta_0}(x)$ and $\mathcal{F}_\theta(x)$ for the input $x$, even when employing the suggested Lipsum-FT regularization, as Lipsum-FT specifically regulates inner product values related to text features instead of simply regularizing the image features to be similar to their original values. It is a valid point and, in fact, acts as the primary reason why Lipsum-FT can be effective for robust fine-tuning. To be more specific, Lipsum-FT goes beyond simply reducing feature distortion in all directions, e.g., through a simple FeatKD regularization which minimizes $\hat{\mathcal{R}}_{\texttt{FeatKD}}(\theta) = \|\mathcal{F}_\theta(x) - \mathcal{F}_{\theta_0}(x)\|_2^2$. Instead, it explicitly addresses feature distortion that negatively impacts the pre-trained vision-language connection, quantified by inner product values related to text features. In other words, *Lipsum-FT allows a degree of flexibility for the feature vector to undergo*

**Table 9: Ablation results on loss function and text guidance.** It supplements Table 3.

**(a)** ImageNet results.

| Method | Loss function
$KLD \to MSE$ | Text guidance
$fixed \to random$ | ImageNet
IN | V2 | R | A | S |
|---|---|---|---|---|---|---|---|
| CAR-FT | | | 83.2 | 73.0 | 71.3 | 43.7 | 49.5 |
| CAR-FT$_{MSE}$ | ✓ | | 83.2 (+0.0) | 73.2 (+0.2) | 74.4 (+3.1) | 49.2 (+5.5) | 51.1 (+1.6) |
| Lipsum-FT$_{KLD}$ | | ✓ | 83.4 (+0.2) | 73.7 (+0.7) | 75.6 (+4.3) | 48.4 (+4.7) | 51.2 (+1.7) |
| Lipsum-FT | ✓ | ✓ | 83.3 (+0.1) | 73.6 (+0.6) | 75.9 (+4.6) | 49.9 (+6.2) | 51.4 (+1.9) |

**(b)** Numbers with standard deviations.

| Method | DomainNet
R | P | C | I | S | ImageNet
IN | V2 | R | A | S |
|---|---|---|---|---|---|---|---|---|---|---|
| CAR-FT | 88.9±0.0 | 64.4±0.1 | 65.8±0.2 | 43.3±0.2 | 53.2±0.5 | 83.2±0.0 | 73.0±0.2 | 71.3±0.3 | 43.7±0.2 | 49.5±0.2 |
| CAR-FT$_{MSE}$ | 88.9±0.1 | 65.7±0.2 | 67.1±0.3 | 44.5±0.2 | 55.2±0.3 | 83.2±0.0 | 73.2±0.1 | 74.4±0.2 | 49.2±0.3 | 51.1±0.3 |
| Lipsum-FT$_{KLD}$ | 89.0±0.0 | 66.0±0.1 | 67.7±0.3 | 45.8±0.2 | 55.7±0.3 | 83.4±0.0 | 73.7±0.1 | 75.6±0.1 | 48.4±0.4 | 51.2±0.1 |
| Lipsum-FT | 89.0±0.0 | 66.3±0.2 | 68.0±0.0 | 46.0±0.2 | 56.2±0.2 | 83.3±0.0 | 73.6±0.1 | 75.9±0.1 | 49.9±0.3 | 51.4±0.1 |

*changes after fine-tuning, as long as it does not compromise the pre-trained vision-language connection.* Given that the linear probing baseline, which freezes features entirely, falls short of delivering satisfactory downstream performance, it becomes evident that introducing a degree of feature distortion is necessary for improved results in downstream tasks. This poses a key challenge in robust fine-tuning, specifically in deciding how to control the level of distortion within an acceptable range. In response to this challenge, our Lipsum-FT approach maintains a suitable 'certain level of feature distortion' by retaining the pre-trained vision-language connection, effectively addressing both reference and distribution shift performance in downstream tasks in the context of robust fine-tuning. Conducting an empirical investigation, we tested the aforementioned FeatKD method in the ImageNet scenario, yielding classification accuracy of $\{83.1_{\pm0.1}, 73.1_{\pm0.2}, 72.0_{\pm0.1}, 43.4_{\pm0.3}, 49.9_{\pm0.2}\}$ on ImageNet-$\{$IN,V2,R,A,S$\}$. While these results are commendable, they fall short when compared to Lipsum-FT. This underscores the effectiveness of the Lipsum-FT approach, especially in terms of its flexibility to accommodate feature distortion.

**DomainNet results with different reference data.** We extend our main experiments for the DomainNet scenario by using different domain as reference data for fine-tuning and employing other domains as distribution shift data. It might provide additional insights into the adaptability and robustness of the proposed method. Table 10 summarizes the classification accuracy results for B/16 in four different distribution shift scenarios on DomainNet. It is important to note that we applied the same hyperparameters as the DomainNet-R → DomainNet-$\{$P,C,I,S$\}$ scenario presented in the main text, without any adjustments. The results distinctly indicate that the proposed Lipsum-FT approach consistently outperforms other baseline methods in these scenarios.

**Visualizing energy vectors.** We further depict visualizations of energy vectors in an $M$-dimensional space before and after the fine-tuning process. This aims to provide an intuitive grasp of the distinctions between initial zero-shot and fine-tuned models. Given the high dimensions, such as $M = 80$, we project these vectors into a two-dimensional space using t-SNE (van der Maaten & Hinton, 2008). Specifically, we generate energy vectors for the first 100 data points from each split and visualize them in a two-dimensional space using sklearn.manifold.TSNE with default parameters. Figure 15 clearly demonstrates that Lipsum-FT displays the least distinction in comparison with other baseline methods. It implies that Lipsum-FT effectively alleviate the decline in pre-trained connections between vision and language models.

**Analysis on training costs.** We provide the wall-clock times for the training runs of the fine-tuning methods to offer a clearer understanding of the computational complexity of each method. All training runs occurred in a consistent TPU VM environment with eight TPU-v3 cores. It would be crucial to acknowledge that, even when running the same script on the same VM within a TPU development environment, there may be variations in execution times. Hence, we supply average

**Table 10: Results with varying reference data on DomainNet.** It summarizes the accuracy of various methods for `B/16` in distribution shift scenarios on DomainNet. The split denoted by an asterisk(*) serves as the reference dataset. An underline highlights the top two values in each column, where all values are averaged from the three measurements.

**(a)** `DomainNet-P → DomainNet-{R,C,I,S}`

| Method | R | P* | C | I | S | Average |
|---|---|---|---|---|---|---|
| FT | $74.2_{\pm0.1}$ | $79.8_{\pm0.0}$ | $60.9_{\pm0.4}$ | $38.7_{\pm0.4}$ | $48.5_{\pm0.6}$ | 60.4 |
| EMA | $\underline{78.4}_{\pm0.1}$ | $\underline{80.8}_{\pm0.1}$ | $\underline{65.2}_{\pm0.1}$ | $\underline{42.1}_{\pm0.1}$ | $\underline{54.4}_{\pm0.0}$ | $\underline{64.2}$ |
| L2SP | $74.2_{\pm0.2}$ | $79.8_{\pm0.3}$ | $60.4_{\pm0.9}$ | $38.4_{\pm0.3}$ | $48.0_{\pm0.3}$ | 60.2 |
| KD | $77.2_{\pm0.1}$ | $79.7_{\pm0.2}$ | $62.2_{\pm0.3}$ | $39.4_{\pm0.2}$ | $50.3_{\pm0.4}$ | 61.8 |
| CAR-FT | $76.0_{\pm0.1}$ | $80.1_{\pm0.0}$ | $61.6_{\pm0.4}$ | $40.0_{\pm0.5}$ | $50.1_{\pm0.4}$ | 61.6 |
| Lipsum-FT (ours) | $\underline{78.9}_{\pm0.1}$ | $\underline{80.6}_{\pm0.1}$ | $\underline{64.8}_{\pm0.3}$ | $\underline{43.1}_{\pm0.2}$ | $\underline{53.8}_{\pm0.1}$ | $\underline{64.2}$ |

**(b)** `DomainNet-C → DomainNet-{R,P,I,S}`

| Method | R | P | C* | I | S | Average |
|---|---|---|---|---|---|---|
| FT | $76.2_{\pm0.2}$ | $58.1_{\pm0.3}$ | $79.9_{\pm0.1}$ | $39.8_{\pm0.2}$ | $53.5_{\pm0.2}$ | 61.5 |
| EMA | $\underline{79.9}_{\pm0.1}$ | $\underline{63.2}_{\pm0.2}$ | $\underline{81.4}_{\pm0.1}$ | $\underline{44.1}_{\pm0.1}$ | $\underline{58.9}_{\pm0.2}$ | $\underline{65.5}$ |
| L2SP | $76.0_{\pm0.1}$ | $58.1_{\pm0.4}$ | $79.7_{\pm0.1}$ | $40.1_{\pm0.5}$ | $53.5_{\pm0.5}$ | 61.5 |
| KD | $78.7_{\pm0.2}$ | $61.1_{\pm0.3}$ | $79.6_{\pm0.0}$ | $41.9_{\pm0.1}$ | $55.2_{\pm0.4}$ | 63.3 |
| CAR-FT | $78.1_{\pm0.1}$ | $60.5_{\pm0.2}$ | $80.7_{\pm0.2}$ | $43.0_{\pm0.1}$ | $55.4_{\pm0.2}$ | 63.5 |
| Lipsum-FT (ours) | $\underline{80.8}_{\pm0.0}$ | $\underline{63.8}_{\pm0.2}$ | $\underline{81.0}_{\pm0.1}$ | $\underline{45.8}_{\pm0.1}$ | $\underline{58.7}_{\pm0.0}$ | $\underline{66.0}$ |

**(c)** `DomainNet-I → DomainNet-{R,P,C,S}`

| Method | R | P | C | I* | S | Average |
|---|---|---|---|---|---|---|
| FT | $77.3_{\pm0.4}$ | $60.7_{\pm1.3}$ | $61.0_{\pm1.0}$ | $57.5_{\pm0.5}$ | $50.2_{\pm1.6}$ | 61.3 |
| EMA | $\underline{79.3}_{\pm0.2}$ | $\underline{62.3}_{\pm0.1}$ | $\underline{63.5}_{\pm0.1}$ | $\underline{58.7}_{\pm0.1}$ | $\underline{52.6}_{\pm0.2}$ | $\underline{63.3}$ |
| L2SP | $77.4_{\pm0.6}$ | $60.2_{\pm0.9}$ | $61.2_{\pm1.6}$ | $57.5_{\pm0.6}$ | $49.9_{\pm1.9}$ | 61.2 |
| KD | $78.0_{\pm1.0}$ | $60.5_{\pm1.8}$ | $60.6_{\pm1.9}$ | $57.1_{\pm0.9}$ | $48.7_{\pm2.4}$ | 61.0 |
| CAR-FT | $76.9_{\pm0.3}$ | $59.2_{\pm0.2}$ | $59.6_{\pm0.7}$ | $57.4_{\pm0.3}$ | $48.8_{\pm1.0}$ | 60.4 |
| Lipsum-FT (ours) | $\underline{79.9}_{\pm0.1}$ | $\underline{62.9}_{\pm0.2}$ | $\underline{63.9}_{\pm0.3}$ | $\underline{58.8}_{\pm0.2}$ | $\underline{53.4}_{\pm0.3}$ | $\underline{63.8}$ |

**(d)** `DomainNet-S → DomainNet-{R,P,C,I}`

| Method | R | P | C | I | S* | Average |
|---|---|---|---|---|---|---|
| FT | $71.5_{\pm0.4}$ | $56.0_{\pm0.2}$ | $65.4_{\pm0.7}$ | $32.2_{\pm1.0}$ | $71.6_{\pm0.2}$ | 59.3 |
| EMA | $\underline{76.4}_{\pm1.3}$ | $\underline{61.4}_{\pm1.5}$ | $\underline{68.7}_{\pm1.0}$ | $\underline{37.4}_{\pm1.1}$ | $\underline{73.5}_{\pm0.4}$ | $\underline{63.5}$ |
| L2SP | $71.5_{\pm0.2}$ | $55.8_{\pm0.1}$ | $65.7_{\pm0.3}$ | $32.8_{\pm0.5}$ | $71.7_{\pm0.1}$ | 59.5 |
| KD | $74.9_{\pm0.1}$ | $59.7_{\pm0.2}$ | $66.7_{\pm0.4}$ | $34.7_{\pm0.4}$ | $71.2_{\pm0.1}$ | 61.4 |
| CAR-FT | $73.6_{\pm0.2}$ | $57.3_{\pm0.2}$ | $66.2_{\pm0.0}$ | $35.2_{\pm0.3}$ | $72.0_{\pm0.2}$ | 60.9 |
| Lipsum-FT (ours) | $\underline{78.0}_{\pm0.1}$ | $\underline{62.4}_{\pm0.2}$ | $\underline{68.3}_{\pm0.1}$ | $\underline{40.1}_{\pm0.5}$ | $\underline{72.4}_{\pm0.2}$ | $\underline{64.2}$ |

and standard deviation values from five runs, effectively fulfilling the purpose of comparing overall training costs. The following summarizes the wall-clock times for training runs on the ImageNet scenario with `50000` training steps: `FT` took $175 \pm 8$ minutes, `EMA` took $175 \pm 11$ minutes, `L2SP` took $177 \pm 8$ minutes, `KD` took $209 \pm 11$ minutes, `CAR-FT` took $207 \pm 14$ minutes, and `Lipsum-FT` took $211 \pm 18$ minutes, Overall, `KD`, `CAR-FT`, and `Lipsum-FT` approaches, requiring an additional forward pass compared to FT, lead to an approximately 20% increase in training time. It would be worth mentioning that `Lipsum-FT` requires virtually no cost over `KD` and `CAR-FT`.

**Results for generalized zero-shot learning.** The focus of the paper is addressing distribution shift scenarios under fixed classes, as opposed to evaluating the zero-shot capability across both *seen* and *unseen* classes. However, one may contemplate situations related to generalized zero-shot learning.

**Table 11: Generalized zero-shot learning results on CUB.** It summarizes the classification accuracy for seen classes (S), unseen classes (U), and their harmonic mean (H). Our outcomes are averaged from three measurements, and the corresponding standard deviation values are provided. [†]These are the baseline results adopted from Wang et al. (2023).

| Method | S | U | H |
|---|---|---|---|
| Zero-shot | 55.3 | 52.5 | 53.9 |
| FT | $83.1_{\pm0.3}$ | $33.6_{\pm0.8}$ | $47.8_{\pm0.5}$ |
| CAR-FT | $\underline{85.2}_{\pm0.1}$ | $44.6_{\pm0.6}$ | $58.5_{\pm0.5}$ |
| Lipsum-FT | $82.6_{\pm0.2}$ | $48.5_{\pm0.5}$ | $61.1_{\pm0.3}$ |
| [†]Zero-shot | 54.8 | 55.2 | 55.0 |
| [†]CoOp (Zhou et al., 2022b) | 63.8 | 49.2 | 55.6 |
| [†]CoOp + SHIP (Wang et al., 2023) | 58.9 | 55.3 | 57.1 |

To address this, we further present outcomes for generalized zero-shot learning, using the Caltech-UCSD Birds 200 (CUB) dataset[2]. In accordance with the official codebase of Xian et al. (2018)[3], we partitioned the original set of 200 classes into 150 classes designated as seen and 50 classes as unseen, the latter being excluded from the training set. Due to the incapacity of the conventional softmax classifier to handle unseen classes, we opted for fixed zero-shot head weights derived from the pre-trained text model of CLIP. To elaborate, we commenced by fine-tuning the vision model, incorporating the fixed zero-shot head weights, using the training data for the 150 seen classes. Subsequently, we evaluated the performance of the fine-tuned vision model, equipped with the fixed zero-shot head weights, on the test data comprising 150 seen classes and 50 unseen classes. Table 11 summarizes the results averaged from three trials; S represents the accuracy for 150 seen classes, U represents the accuracy for 50 unseen classes, and H denotes the harmonic mean of these values. In comparison to FT, Lipsum-FT demonstrates enhanced proficiency in maintaining the original superior performance of the zero-shot model for unseen classes (specifically, Lipsum-FT drops from 52.5 to 48.5, while FT drops 52.5 to 33.6). This serves as clear evidence of the effectiveness of the Lipsum-FT regularization in preserving the pre-trained vision-language connection. Notably, despite not originally targeting a generalized zero-shot learning scenario, Lipsum-FT achieves a performance level that is on par with generalized zero-shot learning methodologies.

---

[2] https://www.vision.caltech.edu/datasets/cub_200_2011/
[3] https://github.com/Abhipanda4/Feature-Generating-Networks/tree/master

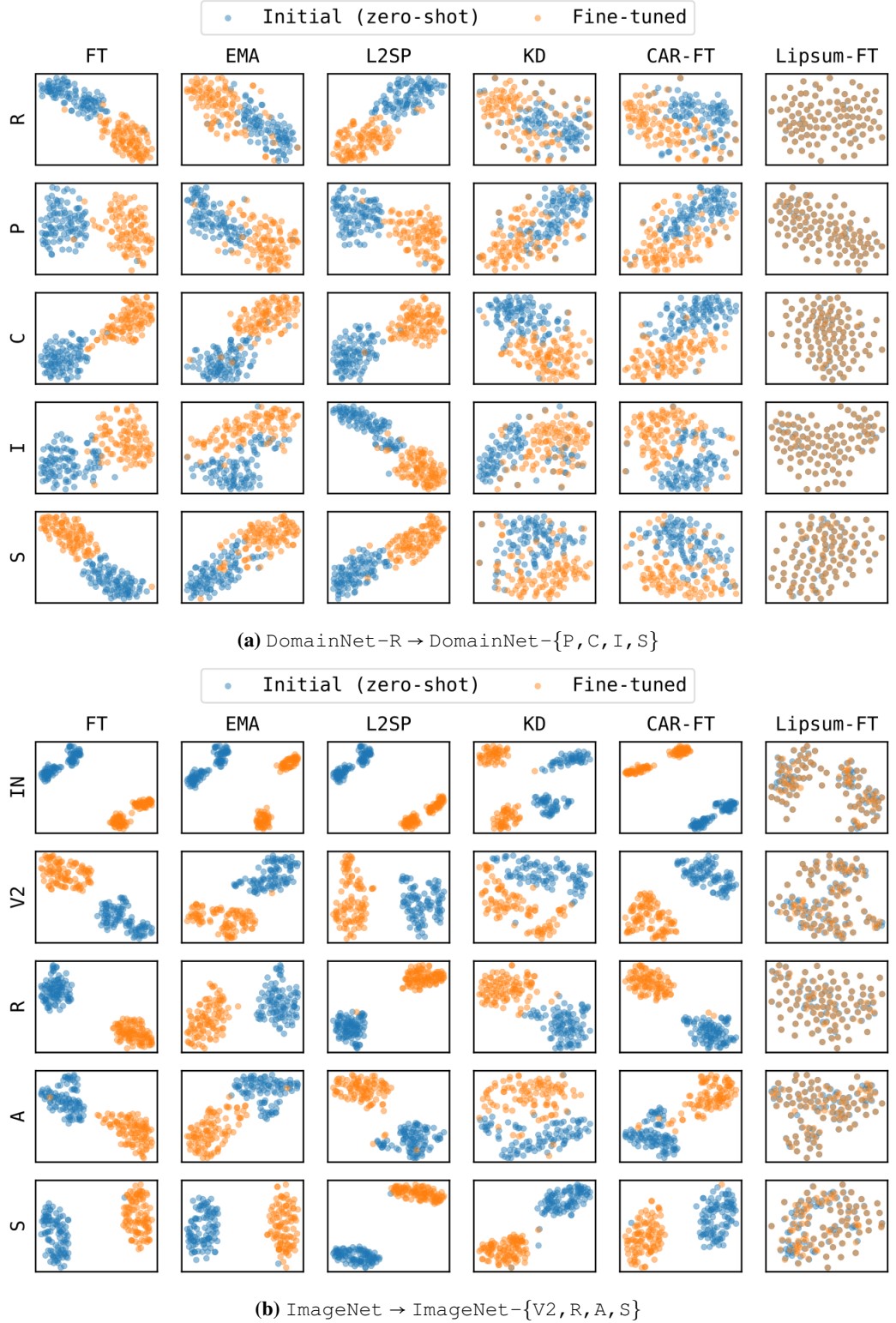

**(a)** `DomainNet-R → DomainNet-{P,C,I,S}`

**(b)** `ImageNet → ImageNet-{V2,R,A,S}`

**Figure 15: Two-dimensional t-SNE visualizations of energy vectors.** It depicts two-dimensional t-SNE representations of $M$-dimensional energy vectors for the initial 100 data points in each data split, both before (represented as $v_{\theta_0,\phi}$ with blue markers) and after fine-tuning (represented as $v_{\theta,\phi}$ with orange markers).

## C EXPERIMENTAL DETAILS

The code for the main experiments is built on `JAX` (Bradbury et al., 2018), `Flax` (Babuschkin et al., 2020), `Optax` (Babuschkin et al., 2020), `TensorFlow Datasets` (Abadi et al., 2015), and `Transformers` (Wolf et al., 2020) libraries, which are available under the Apache-2.0 licence.[4] All experiments are carried out using eight TPUv2 or TPUv3 cores, supported by TPU Research Cloud.[5] Code is available at `https://github.com/cs-giung/lipsum-ft`.

**Pre-trained weights.** Throughout the experiments, we employed the pre-trained weights provided by OpenAI. These weights can be accessed publicly through the following URLs:

- `B/32`: `https://huggingface.co/openai/clip-vit-base-patch32`
- `B/16`: `https://huggingface.co/openai/clip-vit-base-patch16`
- `L/14`: `https://huggingface.co/openai/clip-vit-large-patch14`

**Datasets.** We employed the following datasets in this work, where Figure 16 visually depicts example images from each dataset: **(1) ImageNet** (`IN`; Russakovsky et al., 2015) consists of images from 1,000 categories. Due to its significance in the field of computer vision, several related datasets have been introduced: ImageNet-V2 (`V2`; Recht et al., 2019), ImageNet-Rendition (`R`; Hendrycks et al., 2021a), ImageNet-A (`A`; Hendrycks et al., 2021b), and ImageNet-Sketch (`S`; Wang et al., 2019). These datasets are considered as *natural distribution shifts* of the ImageNet dataset (Taori et al., 2020), and it is a common practice to evaluate the robustness of models fine-tuned on ImageNet using these datasets (Taori et al., 2020; Radford et al., 2021; Wortsman et al., 2022a;b; Mao et al., 2023; Tian et al., 2023). **(2) DomainNet** (Peng et al., 2019) comprises images categorized into 345 classes from six distinct domains, which include photos (`R`), paintings (`P`), clipart (`C`), infographics (`I`), sketches (`S`), and quickdraw (`Q`). We excluded the quickdraw data from all experiments since obtaining meaningful results, either before or after fine-tuning, is difficult due to its extreme simplicity (to be more precise, the classification accuracy remains around 5% in both cases). To address out-of-distribution generalization rather than domain generalization, we fine-tuned models solely on `R` data and then evaluated them on all domains.

---

[4]`https://www.apache.org/licenses/LICENSE-2.0`
[5]`https://sites.research.google/trc/about/`

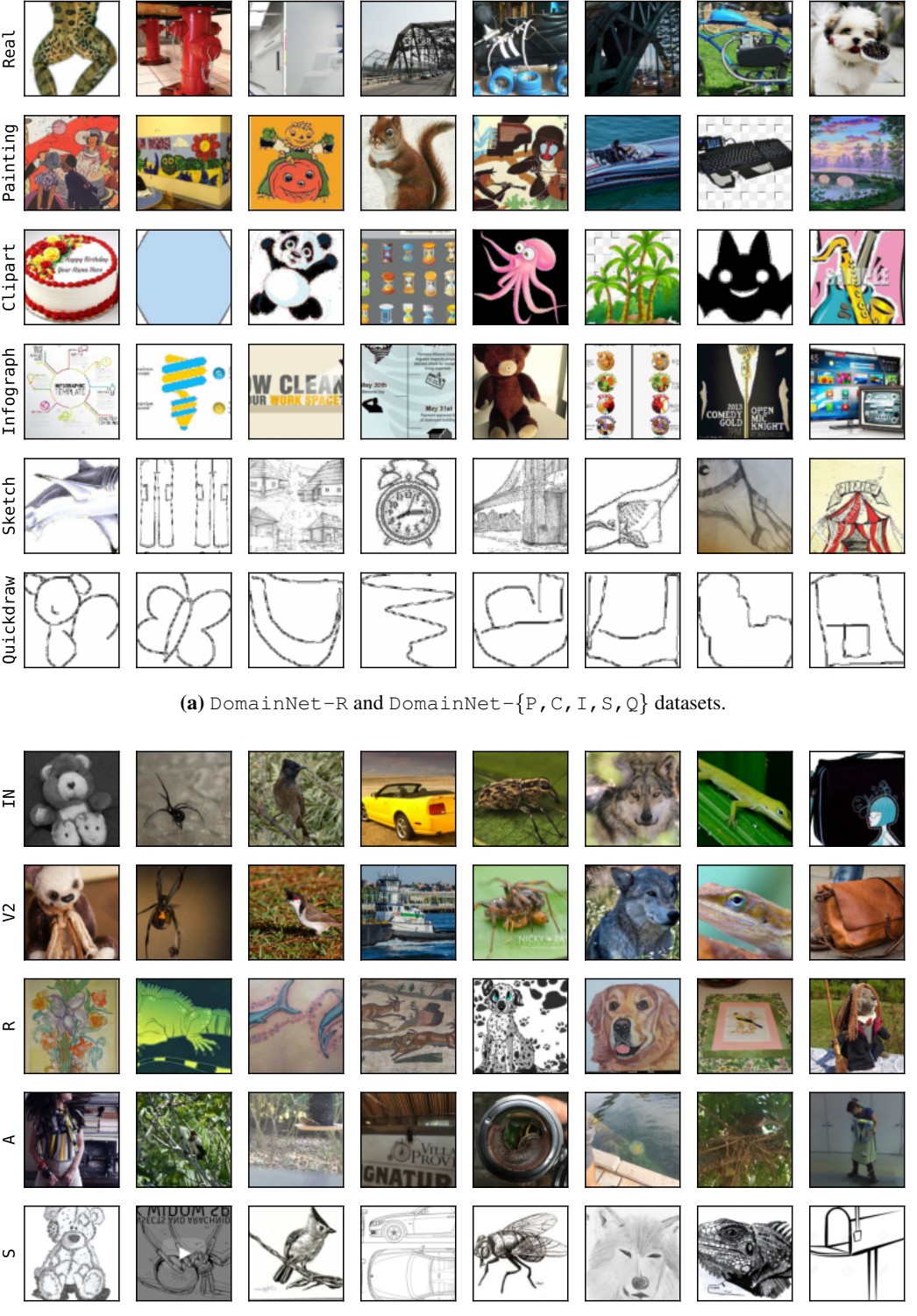

**(a)** `DomainNet-R` and `DomainNet-{P,C,I,S,Q}` datasets.

**(b)** `ImageNet` and `ImageNet-{V2,R,A,S}` datasets.

**Figure 16: Example images from datasets.** It shows 10 randomly chosen images from (a) DomainNet datasets and (b) ImageNet datasets, respectively.

