# OpenReview forum: "Lipsum-FT: Robust Fine-Tuning of Zero-Shot Models Using Random Text Guidance"
_ICLR.cc/2024/Conference — ICLR 2024 poster_

### Official Review · Reviewer_Q7od · 2023-10-29

**Soundness:** 3 good
**Presentation:** 3 good
**Contribution:** 2 fair
**Rating:** 6
**Confidence:** 5

**Summary:**

The paper introduces a method named LIPSUM-FT for zero-shot robust fine-tuning. Specifically, the authors, by observing the relationship between accuracy and energy gap under distribution shift, propose enhancing the accuracy of fine-tuned pre-trained models under distribution shifts by reducing the energy gap. The experimental results demonstrate that the method proposed in this paper can effectively improve the performance of CLIP during zero-shot fine-tuning.

**Strengths:**

1. The authors explain the performance degradation of fine-tuned pre-trained models under distribution shifts from the perspective of energy models.

2. The proposed LIPSUM-FT effectively enhances the robustness of fine-tuned pre-training.

3. Ablation studies indicate that LIPSUM-FT is insensitive to random token lengths and token quantities.

**Weaknesses:**

1. Utilizing energy models to explain the fine-tuning of pre-trained models seems not to be essential. As per my understanding, the objective of the method in this paper as well as related methods ([1,2,3], etc.) is to reduce the difference in features extracted by the models before and after fine-tuning.

2. The authors claim that the text used is randomly generated, but it appears from the code in the supplementary material that tokens are sampled from the openai_imagenet_template. According to CAR-FT, using all templates as text input also yields good performance. What then is the significance of random token sampling in this scenario?

3. It is suggested that the authors provide a brief introduction to energy models in the related work section.
In Figure 1, it is not mentioned which points different learning rates in the left graph and different steps in the right graph correspond to.

[1] Context-aware robust fine-tuning.

[2] Fine-tuning can cripple your foundation model; preserving features may be the solution.

[3] Robust fine-tuning of zero-shot models.

**Questions:**

The authors claim that the text tokens are randomly generated. What are the specific rules for generation?

---

> ### Author Response · Authors · 2023-11-11
> **Individual Response (1)**
>
> We are grateful for your thorough review of our paper. Initially, we would like to offer the following responses, along with the list of planned action items.
>
> ===
>
> > Utilizing energy models to explain the fine-tuning of pre-trained models seems not to be essential. As per my understanding, the objective of the method in this paper as well as related methods ([1,2,3], etc.) is to reduce the difference in features extracted by the models before and after fine-tuning.
>
> In the main text, we have covered the seminal work conducted by Kumar et al. (2022), where they posited that fine-tuning has the potential to modify the features acquired in pre-training. Furthermore, we explored practical methods for robust fine-tuning, including the contributions of Wortsman et al. (2022b) and Mao et al. (2022) you mentioned. The work by Mukhoti et al. (2023) also aligns with the concept of robust fine-tuning we discussed (note that Mukhoti et al. (2023) is contemporaneous, as outlined in the [FAQ for Reviewers](https://iclr.cc/Conferences/2024/ReviewerGuide)).
>
> The idea you mentioned, which involves minimizing the discrepancy in features extracted by models before and after fine-tuning, represents a fundamental principle shared by various robust fine-tuning methodologies we discussed (e.g., weight-space regularization via EMA and L2SP, function-space regularization via KD and CAR-FT, and post-hoc weight-space regularization via WiSE and TPGM). Our work is significant in interpreting the success of these robust fine-tuning methodologies in the context of fine-tuning vision-language models based on a joint energy-based model (Section 4.2). Furthermore, we propose a novel approach called Lipsum-FT, building on this understanding (Section 5).
>
> Ensuring that the vision model maintains the original inner products with features generated by the language model is our distinctive idea and undeniably possesses novelty. It is worth quoting the following by [Michael Black](https://medium.com/@black_51980/novelty-in-science-8f1fd1a0a143): "If it is easy to explain and obvious in hindsight, this in no way diminishes the creativity (and novelty) of the idea."
>
> > The authors claim that the text used is randomly generated, but it appears from the code in the supplementary material that tokens are sampled from the openai_imagenet_template. According to CAR-FT, using all templates as text input also yields good performance. What then is the significance of random token sampling in this scenario?
>
> > The authors claim that the text tokens are randomly generated. What are the specific rules for generation?
>
> In the main text, we employ the zero-shot classification head, obtained from the `open_ai_template`, as the initialization for all fine-tuning methods. As a result, the `open_ai_template` present in the code is exclusively utilized for constructing the zero-shot classification head and is not involved in generating random texts for Lipsum-FT. The following lines of code, responsible for randomly generating text tokens using a vocabulary of size 49406, would clarify your concerns regarding the generation of random text:
> ```
>             txts = [jnp.array(
>                 [49406,] + [0,] * config.token_length + [49407,]).at[
>                     1:1+config.token_length
>                 ].set(jax.random.randint(
>                     rngs[iii], (config.token_length,), minval=0, maxval=49406))
>                 for iii in range(config.token_k_ways)]
> ```
>
> > It is suggested that the authors provide a brief introduction to energy models in the related work section.
>
> Thank you for your valuable suggestion! Including a brief introduction to energy-based models in the related work section is a great idea. It would provide readers with a better context for understanding our research. __We appreciate your input and will supplement the Appendix A section (Action Item #1).__
>
> > In Figure 1, it is not mentioned which points different learning rates in the left graph and different steps in the right graph correspond to.
>
> While we mentioned in the main text that an increase in learning rates or additional training steps results in upward and leftward movement, it is advisable, as you recommended, to provide explicit specifications on plots to improve readability. __We will accordingly revise Figures 1 and 6 (Action Item #2).__ Thank you for your constructive feedback!
>
> ===
>
> __We will share the upcoming action items once they are completed. If there are any remaining concerns, please let us know. Otherwise, we would like to ask you to raise your assessment accordingly.__
> * __(Action Item #1) An additional paragraph regarding energy-based models in the related work section.__
> * __(Action Item #2) Improving readability of Figures 1 and 6.__

---

> > ### Comment · Reviewer_Q7od · 2023-11-12
> >
> > The author's response has clarified some of our confusions regarding the generation of random text, and we have raised the score to 6. We hope the author will update the Action Items in the paper or the appendix。

---

> > > ### Author Response · Authors · 2023-11-12
> > > **Individual Response (2)**
> > >
> > > We appreciate your positive reassessment! We posted the first revised version, encompassing content associated with Action Items #1 and #2 you mentioned. Furthermore, we would encourage you to take a look at the supplementary materials for the action items provided by other reviewers, hoping they also align with your satisfaction.

---

### Official Review · Reviewer_26HN · 2023-10-31

**Soundness:** 3 good
**Presentation:** 2 fair
**Contribution:** 2 fair
**Rating:** 6
**Confidence:** 3

**Summary:**

This paper focuses on the problem of robust fine-tuning a large-scale vision-language pre-trained model, which is expected to obtain enhanced downstream performance while maintaining its accuracy across distribution shifts on zero-shot tasks. By investigating the behavior under the perspectives of feature distortion theory and joint energy-based models, the authors propose a robust fine-tuning algorithm Lipsum-FT. Experimental results on DomainNet and ImageNet proves effectiveness of their proposed method.

**Strengths:**

1. This paper focuses on the problem of how to maintain the performance of a zero-shot model on distribution shift data while improving its performance on reference data during fine-tuning, which is an important and valuable topic.
2. The proposed method is easy to realize since it simply introduces an extra regularization term.
3. The idea of utilize the language model to construct regularization on the vision model is interesting.
4. The English writing is good and I do not find obvious grammar errors or typos.

**Weaknesses:**

1. I am not sure whether the novelty of this paper can be regarded as significant. It just introduces a regularization item to make the vision model keep the original inner products with features generated by the language model after fine-tuning.
2. The illustration organization of this paper is not clear enough. Therefore, even though the key idea is not so complicated, I find it difficult to understand the viewpoint quickly.

**Questions:**

1. As the main contribution of this paper is to utilize the regularization term in (7) to minimize the energy gap in (6), please explain why the energy gap is chosen for improving the robustness of zero-shot model fine-tuning. Why is it defined to be the squared difference of two inner products as in (6)?
2. Could the authors give more details about how the tokens $\mathbf{t}$ are generated? The authors just assert that they are generated randomly from the vocabulary. I also want to know what types of texts are used for constructing such regularization and to what extent it covers the common used semantic information.
3. How much extra computation are introduced by such a regularization term?

**Details Of Ethics Concerns:**

I have no ethics concerns about this paper.

---

> ### Author Response · Authors · 2023-11-11
> **Individual Response (1a)**
>
> We appreciate the comprehensive review of our paper. At the outset, we would like to present the following responses, along with a list of intended action items.
>
> ===
>
> > I am not sure whether the novelty of this paper can be regarded as significant. It just introduces a regularization item to make the vision model keep the original inner products with features generated by the language model after fine-tuning.
>
> We acknowledge your perspective but hold a distinct view on novelty. Our method extends established concepts in robust fine-tuning, where existing approaches share the fundamental principle of mitigating the fine-tuned model from deviating significantly from the pre-trained model.
>
> As an illustration, Wortsman et al. (2022b) achieve robust fine-tuning by (A) empirically observing the fine-tuned model residing in the same pre-trained basin (as seen in Neyshabur et al., 2020) and (B) implementing a straightforward weight averaging strategy (referencing Izmailov et al., 2018; Wortsman et al., 2022a). Similarly, our work introduces a novel (A) exploration using feature distortion theory and a joint energy-based model (referencing Kumar et al., 2022; Grathwohl et al., 2019) and (B) feature regularization strategy utilizing random text in the context of fine-tuning vision-language pre-trained models.
>
> Ensuring that the vision model maintains the original inner products with features generated by the language model is our distinctive idea and undeniably possesses novelty. It is worth quoting the following by [Michael Black](https://medium.com/@black_51980/novelty-in-science-8f1fd1a0a143): "If it is easy to explain and obvious in hindsight, this in no way diminishes the creativity (and novelty) of the idea."
>
> > The illustration organization of this paper is not clear enough. Therefore, even though the key idea is not so complicated, I find it difficult to understand the viewpoint quickly.
>
> Thank you for your valuable feedback! __We plan to draw a conceptual figure that showcases the overall idea of this work, as we think it could address this concern (Action Item #3).__ It will undoubtedly enhance the readability of the paper and aid readers in better understanding. We appreciate once again for the constructive feedback.
>
> > As the main contribution of this paper is to utilize the regularization term in (7) to minimize the energy gap in (6), please explain why the energy gap is chosen for improving the robustness of zero-shot model fine-tuning. Why is it defined to be the squared difference of two inner products as in (6)?
>
> The foundational assumption guiding energy gap minimization is that current robust fine-tuning techniques alleviate the decline in "pre-trained connections" between vision and language models. The term "pre-trained connections" carries some ambiguity, and one of our primary contributions involves quantifying it based on the joint energy-based model. Consequently, the notion of "weakening pre-trained connections" can be characterized as a degradation in discriminative modeling between the two, evident in an increased energy gap. Accordingly, Lipsum-FT, our proposed method, employs a regularization term to minimize the energy gap, effectively accomplishing the objective of robust fine-tuning.

---

> > ### Author Response · Authors · 2023-11-11
> > **Individual Response (1b)**
> >
> > > Could the authors give more details about how the tokens are generated? The authors just assert that they are generated randomly from the vocabulary. I also want to know what types of texts are used for constructing such regularization and to what extent it covers the common used semantic information.
> >
> > The following lines of code, responsible for randomly generating text tokens using a vocabulary of size 49406, would clarify your concerns regarding the generation of random text:
> > ```
> >             txts = [jnp.array(
> >                 [49406,] + [0,] * config.token_length + [49407,]).at[
> >                     1:1+config.token_length
> >                 ].set(jax.random.randint(
> >                     rngs[iii], (config.token_length,), minval=0, maxval=49406))
> >                 for iii in range(config.token_k_ways)]
> > ```
> > __Providing concrete examples of the text generated is a great idea to make the paper solid (Action Item #4).__ Thank you for the constructive feedback! The actual random texts, with a length of 8, involved in the training of Lipsum-FT look like the following. Due to their nature of being generated as sequences of completely random text tokens, it is challenging for them to contain any specific semantic meaning. However, surprisingly, Lipsum-FT empirically demonstrates superior performance using these random texts. Consequently, exploring a novel text guidance strategy customized for robust fine-tuning, rather than relying on random text, could be a fascinating direction for future research.
> > ```
> > [49406  4665 34356 35720 16917 40783 42215   971 12205 49407]
> > ['<|startoftext|>', 'wy', 'prism', 'ayat', 'harold', 'ghead', 'seagulls', 'tt', 'rapid', '<|endoftext|>']
> > [49406 13212 20894 23105 19637 36399 36513 48145 20958 49407]
> > ['<|startoftext|>', 'locks', 'sasha', 'tte', 'beingsalmankhan', 'aras', 'viswas', 'tote', 'aftermath', '<|endoftext|>']
> > [49406 38664  8670  2313  1162 43327  1281 48089  4326 49407]
> > ['<|startoftext|>', 'amat', 'shield', 'league', 'gam', 'decree', 'air', 'protested', 'dog', '<|endoftext|>']
> > ```
> >
> > > How much extra computation are introduced by such a regularization term?
> >
> > Thank you for underscoring the significant concern regarding additional training costs, especially when utilizing large models. In the provided code, Lipsum-FT's implementation entails extracting the text model's output for random text at each iteration, necessitating the forward pass of the text model. However, the output of a fixed text model for entirely random text can be precomputed in practice (similar to CAR-FT). The distinction lies in that while CAR-FT employs 80 guidance vectors, Lipsum-FT, for example, may select 80 from a pool of 10,000 guidance vectors for each iteration. Consequently, we can implement Lipsum-FT using the same computational resources as KD and CAR-FT. __We will provide additional analysis on training costs via wall-clock time, the most practical way to measure the cost (Action Item #5).__
> >
> > ===
> >
> > __We will share the upcoming action items once they are completed. If there are any remaining concerns, please let us know. Otherwise, we would like to ask you to raise your assessment accordingly.__
> > * __(Action Item #3) A conceptual figure demonstrating the overall idea of the work.__
> > * __(Action Item #4) Showing example text randomly generated for Lipsum-FT.__
> > * __(Action Item #5) An additional analysis on training costs.__

---

> > ### Comment · Reviewer_26HN · 2023-11-19
> >
> > Thanks to the authors for their detailed responses, which have resorted some of my concerns. The newly added figure for illustration of the whole framework makes the idea obviously more clear and I suggest the authors add it to the main part of the paper for clarity. I also suggest the authors describe their idea in a clearer and simpler way since I agree with the viewpoint that a good idea is unnecessary to be intricate.
> > However, I need the authors give more explanation to my first question about their motivation of designing such form of regularization. It's notable that two image features given by the vision model may be quite different even if they give similar inner products with a text feature given by the language model, especially when the inner products are close to 0. In this case, the regularization cannot guarantee the two image features to be similar. Is it reasonable for the robust fine-tuning task? Why can the proposed method give better performance than simply regularizing the image features to be similar to their original values?

---

> ### Author Response · Authors · 2023-11-19
>
> > Thanks to the authors for their detailed responses, which have resorted some of my concerns. The newly added figure for illustration of the whole framework makes the idea obviously more clear and I suggest the authors add it to the main part of the paper for clarity. I also suggest the authors describe their idea in a clearer and simpler way since I agree with the viewpoint that a good idea is unnecessary to be intricate.
>
> We are glad that our revisions have met with your satisfaction! Following your suggestion, we will place the illustration depicting the overall idea at the beginning of the text, which will undoubtedly aid readers' understanding. Thank you once again for your constructive feedback!
>
> > However, I need the authors give more explanation to my first question about their motivation of designing such form of regularization. It's notable that two image features given by the vision model may be quite different even if they give similar inner products with a text feature given by the language model, especially when the inner products are close to 0. In this case, the regularization cannot guarantee the two image features to be similar. Is it reasonable for the robust fine-tuning task? Why can the proposed method give better performance than simply regularizing the image features to be similar to their original values?
>
> You pointed out that a distinction might exist between $\mathcal{F}\_{\theta\_{0}}(x)$ and $\mathcal{F}\_{\theta}(x)$ for the input $x$, even when employing the suggested Lipsum-FT regularization, as Lipsum-FT specifically regulates inner product values related to text features instead of "simply regularizing the image features to be similar to their original values." It is a valid point and, in fact, acts as the primary reason why Lipsum-FT can be effective for robust fine-tuning.
>
> To be more specific, Lipsum-FT goes beyond simply reducing feature distortion in all directions (e.g., through the simple regularization like minimizing $||\mathcal{F}\_{\theta}(x) - \mathcal{F}\_{\theta\_{0}}(x)||\_2^2$). Instead, it explicitly addresses feature distortion that negatively impacts the pre-trained vision-language connection, quantified by inner product values related to text features. In other words, _Lipsum-FT allows a degree of flexibility for the feature vector to undergo changes after fine-tuning_ (i.e., "a distinction might exist between $\mathcal{F}\_{\theta\_{0}}(x)$ and $\mathcal{F}\_{\theta}(x)$"), _as long as it does not compromise the pre-trained vision-language connection_ (i.e., "Lipsum-FT specifically regulates inner product values related to text features"). Considering that a linear probing baseline, which completely freezes features, does not achieve satisfactory downstream performance, it is clear that we should allow a certain level of feature distortion for enhanced performance in downstream tasks, and it constitutes a fundamental challenge in robust fine-tuning—determining how to regulate the extent of distortion within an acceptable range. In response to this challenge, our Lipsum-FT regularization design maintains a suitable "certain level of feature distortion" by retaining the pre-trained vision-language connection, effectively addressing both reference and distribution shift performance in downstream tasks in the context of robust fine-tuning.
>
> Here, we provide additional results for the simple regularization mentioned above, i.e., minimizing $||\mathcal{F}\_{\theta}(x) - \mathcal{F}\_{\theta_{0}}(x)||_2^2$, further underscoring the efficacy of our Lipsum-FT regularization method. Labeling this method FeatKD (Feature Knowledge Distillation, as it employs knowledge distillation on output features), the following table compares it for the ImageNet scenario:
>
> | Method    | IN       | V2       | R        | A        | S        |
> | :-        | :-       | :-       | :-       | :-       | :-       |
> | FT | 82.8±0.1 | 72.6±0.3 | 68.5±0.3 | 39.2±0.3 | 48.0±0.2 |
> | FeatKD    | 83.1±0.1 | 73.1±0.2 | 72.0±0.1 | 43.4±0.3 | 49.9±0.2 |
> | KD        | 83.1±0.1 | 73.1±0.3 | 72.9±0.1 | 42.3±0.4 | 49.9±0.2 |
> | CAR-FT    | 83.2±0.0 | 73.0±0.2 | 71.3±0.3 | 43.7±0.2 | 49.5±0.2 |
> | Lipsum-FT | **83.3**±0.0 | **73.6**±0.1 | **75.9**±0.1 | **49.9**±0.3 | **51.4**±0.1 |
>
> We believe that our response comprehensively addresses your remaining concerns. If there are any outstanding issues, please inform us. The final version of the paper will incorporate the discussion on this matter. We appreciate your valuable insights that make our paper solid!

---

> > ### Comment · Reviewer_26HN · 2023-11-20
> >
> > After reading this response, my concerns have been mostly addressed, and I would like to raise my rating to 6. I hope the authors should make corresponding changes in their final version and describe their motivation in a clearer way to make the paper easier to read.

---

> > > ### Author Response · Authors · 2023-11-20
> > >
> > > We are pleased to hear that you found our revision and response satisfactory. We appreciate your positive assessment, and as you mentioned, the final version of the paper will incorporate all the discussed points in a clear and readable format. Thanks again!

---

### Official Review · Reviewer_2e7P · 2023-11-02

**Soundness:** 2 fair
**Presentation:** 2 fair
**Contribution:** 2 fair
**Rating:** 5
**Confidence:** 4

**Summary:**

This paper proposed a fine-tuning model, Lipsum-FT, by utilizing the language modeling aspect of the vision-language pre-trained models for zero-shot classification.

**Strengths:**

This paper proposed a fine-tuning model, Lipsum-FT, by utilizing the language modeling aspect of the vision-language pre-trained models for zero-shot classification. Lipsum-FT as an incremental algorithm enhances the robustness of zero-shot models by combining language and visual information.

**Weaknesses:**

The motivation is unclear. The authors mention a problem with cross-distribution shifts, but Lipsum-FT is a fine-tuning model.

**Questions:**

What means of four distribution shifts DomainNet-{P, C, I, S}. Figure 1 fails to represent the distribution shifts.

There are too many contents that need to be referred to Appendix. Please reduce some important Appendix contents and put them into the text.

It looks like capturing correlations between images and each language by calculating their inner product vectors in Equation 8. The meaning of Equation 8 wants to express is to match all the language information one by one or to match the current m-th information? If it matches M language information, how much does the algorithm complexity increase? What is the significance of matching with alonely m-th?

Lack of complexity analysis on the Lipsum-FT algorithm and its impact on the experiments.

---

> ### Author Response · Authors · 2023-11-11
> **Individual Response (1)**
>
> We appreciate your efforts in reviewing our paper. Firstly, we would like to provide the following responses, along with a list of planned action items.
>
> ===
>
> > The motivation is unclear. The authors mention a problem with cross-distribution shifts, but Lipsum-FT is a fine-tuning model.
>
> We respectfully differ on the concerns about "unclear motivation." Our study is in line with recent progress in robust fine-tuning research (Radford et al., 2021; Pham et al., 2023; Wortsman et al., 2022b; Mao et al., 2022; Tian et al., 2023), addressing the examination and alleviation of performance trade-offs between reference and distribution shift data during the fine-tuning of large-scale pre-trained models.
>
> > What means of four distribution shifts DomainNet-{P, C, I, S}. Figure 1 fails to represent the distribution shifts.
>
> DomainNet serves as a representative dataset for exploring techniques to manage out-of-distribution data. "DomainNet-{P,C,I,S}" denotes paintings, cliparts, infographics, and sketches, respectively. Detailed descriptions are available in Appendix B.4, and visual examples are provided in Figure 14.
>
> Could you please elaborate further the meaning of the statement "Figure 1 fails to represent the distribution shifts"? Figure 1 depicts the occurrence of a performance trade-off between reference and distribution shift data following the fine-tuning of CLIP-ViT models, as discussed earlier in Radford et al. (2021).
>
> > There are too many contents that need to be referred to Appendix. Please reduce some important Appendix contents and put them into the text.
>
> Due to the substantial number of experiments we conducted, there are numerous instances where readers should refer to the appendix, as you mentioned. While we endeavored to incorporate the main experimental results into the main text, typically placing complementary results in the appendix (e.g., presenting Figure 1 for DomainNet in the main text and Figure 6 for ImageNet in the appendix), there may be results in the appendix that merit inclusion in the main text. If you identify any such results, kindly inform us, and we will make the necessary adjustments to the paper.
>
> > It looks like capturing correlations between images and each language by calculating their inner product vectors in Equation 8. The meaning of Equation 8 wants to express is to match all the language information one by one or to match the current m-th information? If it matches M language information, how much does the algorithm complexity increase? What is the significance of matching with alonely m-th?
>
> The process of computing the inner product between a batched feature vector with a shape of `[batch_size, 768]` and a guidance vector $[\mathcal{G}(\boldsymbol{t_1}),...,\mathcal{G}(\boldsymbol{t_M})]^\top$ with a shape of `[M, 768]` is comparable to the process of computing the inner product between the batched feature vector with a shape of `[batch_size, 768]` and a classification head $\mathbf{W}$ with a shape of `[num_classes, 768]` (`768` here is the dimensionality of the extracted features from the ViT-B/16 model). Consequently, it does not require significant computational cost unless `M` is extremely large.
>
> > Lack of complexity analysis on the Lipsum-FT algorithm and its impact on the experiments.
>
> Thank you for underscoring the significant concern regarding additional training costs, especially when utilizing large models. In the provided code, Lipsum-FT's implementation entails extracting the text model's output for random text at each iteration, necessitating the forward pass of the text model. However, the output of a fixed text model for entirely random text can be precomputed in practice (similar to CAR-FT). The distinction lies in that while CAR-FT employs 80 guidance vectors, Lipsum-FT, for example, may select 80 from a pool of 10,000 guidance vectors for each iteration. Consequently, we can implement Lipsum-FT using the same computational resources as KD and CAR-FT. __We will provide additional analysis on training costs via wall-clock time, the most practical way to measure the cost (Action Item #5).__
>
> ===
>
> __We will share the upcoming action items once they are completed. If there are any remaining concerns, please let us know. Otherwise, we would like to ask you to raise your assessment accordingly.__
> * __(Action Item #5) An additional analysis on training costs.__

---

> ### Comment · Reviewer_2e7P · 2023-11-21
>
> Thanks for authors' efforts. I have another suggestion:
>
> Authors only conduct discussions on the large-scale vision-language model based zero-shot learning (ZSL) methods, while the classical ZSL methods are neglected. I srongly encourage authors to take more classical ZSL methods [a-f] into discussions and comparisons, e.g., on ZSL benchmarks (CUB, SUN, AWA2).
>
> [a] Improving Zero-Shot Generalization for CLIP with Synthesized Prompts. In ICCV, 2023.
>
> [b] Evolving Semantic Prototype Improves Generative Zero-Shot Learning. In ICML, 2023.
>
> [c] Closed-form Sample Probing for Learning Generative Models in Zero-shot Learning. In ICLR, 2022.
>
> [d] HSVA: Hierarchical Semantic-Visual Adaptation for Zero-Shot Learning. In NeurIPS, 2021.
>
> [e] Zero-shot learning by convex combination of semantic embeddings. In ICLR, 2013.
>
> [f] Zero-shot Learning with Semantic Output Codes. In NeurIPS, 2009.

---

> ### Author Response · Authors · 2023-11-21
>
> > Thanks for authors' efforts. I have another suggestion:
> >
> > Authors only conduct discussions on the large-scale vision-language model based zero-shot learning (ZSL) methods, while the classical ZSL methods are neglected. I srongly encourage authors to take more classical ZSL methods [a-f] into discussions and comparisons, e.g., on ZSL benchmarks (CUB, SUN, AWA2).
>
> Following your suggestion, we carried out an additional experiment for generalized zero-shot learning using the CUB dataset. Since the conventional softmax classifier cannot handle unseen classes, we employed fixed zero-shot head weights derived from the pre-trained text model of CLIP. To be more specific, we first fine-tuned the vision model (with the fixed zero-shot head weights) on the training data for 150 seen classes. Subsequently, we assessed the performance of the fine-tuned vision model (with the fixed zero-shot head weights) on the test data that included 150 seen classes and 50 unseen classes.
>
> The following table summarizes the results - S represents the accuracy for 150 seen classes, U represents the accuracy for 50 unseen classes, and H denotes the harmonic mean of these values - averaged from three trials. Additionally, for baseline comparisons, we incorporated CoOp and CoOp + SHIP results from Wang et al. (2023).
>
> | Method    | S          | U          | H          |
> | :-        | :-         | :-         | :-         |
> | Zero-shot | 55.3       | 52.5       | 53.9       |
> | FT        | 83.1 ± 0.3 | 33.6 ± 0.8 | 47.8 ± 0.5 |
> | CAR-FT    | __85.2__ ± 0.1 | 44.6 ± 0.6 | 58.5 ± 0.5 |
> | Lipsum-FT | 82.6 ± 0.2 | __48.5__ ± 0.5 | __61.1__ ± 0.3 |
> ||
> | CoOp | __63.8__ | 49.2 | 55.6 |
> | CoOp + SHIP | 58.9 | __55.3__ | __57.1__ |
>
> Our Lipsum-FT approach is more proficient at maintaining the original superior performance of the zero-shot model for unseen classes (i.e., 52.5 → 48.5, while FT exhibits 52.5 → 33.6). It provides clear evidence of the effectiveness of the Lipsum-FT regularization in preserving the pre-trained vision-language connection. Moreover, due to the nature of fine-tuning, where the parameters of the vision model are directly adjusted for the given training data, we observe a significant improvement in performance for seen classes compared to the prompt learning baselines (i.e., CoOp baselines). As a result, our Lipsum-FT achieved the best performance of H=61.1 across the table.
>
> As the discussion period nears its conclusion, we appreciate your understanding of the time constraints that hinder thorough experiments within the given timeframe. Nonetheless, we are confident that the experimental results we provided for generalized zero-shot learning successfully address your remaining concerns. The final manuscript will include additional discussions and further experiments related to generalized zero-shot learning. Thank you for your time in reviewing our paper.
>
> ---
> Wang et al., 2023, Improving Zero-Shot Generalization for CLIP with Synthesized Prompts.

---

> > ### Comment · Reviewer_2e7P · 2023-11-23
> >
> > I strongly encourage authors to take more classical ZSL methods into discussions and comparison in a individual Table as the Table 3 in [a].
> >
> > [a] Improving Zero-Shot Generalization for CLIP with Synthesized Prompts. In ICCV, 2023.

---

> > > ### Author Response · Authors · 2023-11-23
> > >
> > > We have already showcased the outcomes of our experiments on the CUB dataset, and we assure you that additional discussions and experimental results for the remaining methodologies and datasets will be included in the camera-ready version. Following your suggestion, the incorporation of other ZSL methods using CLIP-ViT-B/16 from Table 3 of Wang et al. (2023) yields the following table. Still, our approach outperforms others as well.
> > >
> > > | Method    | S          | U          | H          |
> > > | :-        | :-         | :-         | :-         |
> > > | Zero-shot | 55.3       | 52.5       | 53.9       |
> > > | FT        | 83.1 ± 0.3 | 33.6 ± 0.8 | 47.8 ± 0.5 |
> > > | CAR-FT    | __85.2__ ± 0.1 | 44.6 ± 0.6 | 58.5 ± 0.5 |
> > > | Lipsum-FT | 82.6 ± 0.2 | __48.5__ ± 0.5 | __61.1__ ± 0.3 |
> > > ||
> > > | CLIP | 54.8 | 55.2 | 55.0 |
> > > | CoOp | 63.8 | 49.2 | 55.6 |
> > > | CoOp + SHIP | 58.9 | __55.3__ | __57.1__ |
> > > | TF-VAEGAN | __84.4__ | 21.1 | 34.0 |
> > > | f-VAEGAN | 82.2 | 22.5 | 35.3 |

---

### Official Review · Reviewer_MuzV · 2023-11-02

**Soundness:** 3 good
**Presentation:** 3 good
**Contribution:** 3 good
**Rating:** 6
**Confidence:** 4

**Summary:**

The paper points out that fine-tuning zero-shot model like CLIP can improve downstream performance, however, the accuracy of the fine-tuned model falls short of the original zero-shot model across distribution shifts. To address the chanllenge of robust fine-tuning, authors first delve into the problem by employing feature distortion theory and joint energy-based models. Subsequently, they introduce a novel robust fine-tuning algorithm called Lipsum-FT. This approach leverages random text guidance as a regularization technique during the fine-tuning process to minimize the change in energy. Authors conduct extensive experiments on two datasets to demonstrate the effectiveness of the proposed approach in addressing distribution shift scenarios.

**Strengths:**

1. This paper investigate the trade-off between the reference and distribution shift data when fine-tuning the zero-shot CLIP model, utilizing feature distortion theory and joint energy-based models as analytical tools.
2. This paper proposes a simple and effective regularization term based on the correlation between the similarity of vision features and text features derived from the fine-tuned model and original model respectively. Specifically, the text tokens are generated randomly.
3. The proposed method outperforms the original zero-shot model and exisiting robust fine-tuning methods in both reference and shift domains, demonstrating its superior performance in handling distribution shifts.

**Weaknesses:**

1. It would be benefical if authors could include visualizations of $v_{\theta, \phi}(x)$ and $v_{\theta_0,\phi}$ to provide an inituitive understanding of the distinctions between the original zero-shot model, the fine-tuned model with exisiting methods, and the fine-tuned model with the proposed emthod.
2. Expanding the experiments to involve various domains as reference data for fine-tuning and using other domains for evaluation would enhance the comprehensiveness of the study. This approach can shed light on the adaptability and robustness of the proposed method in different real-world scenarios.

**Questions:**

1. It remains unclear whether there exists a weight for incorporating the regularization term proposed in Eq. (7) into the loss function. In Appendix B.2, authors have discussed existing methods that involve weights related to the regularization term, such as $\lambda_{L2SP}$ in Eq. (10), $\lambda_{KD}$ in Eq. (12), and $\lambda_{CAR-FT}$ in Eq. (12). Authors have also detailed how to select these hyperparameters. However, there seems no mention of how the weight for the proposed method is determined.
2. The precision of the standard deviation values in Table 2(b) should be improved. The values of NLL are presented accurately to two decimal places, whereas the standard deviation values are limited to only one decimal place, with many of them showing as 0.0. Ensuring consistent precision in reporting these values would enhance the clarity and reliability of the results.
3. There may be a typo in the gradient descent update rule presented in Eq. (9). It should be as follows: $\theta_t = \theta_{t-1} - \eta \nabla_\theta L_{CE}(\theta)$. It's advisable for the authors to thoroughly review other equations to ensure they are accurately represented.
4.It would be interesting to know if the proposed method is applicable to various fine-tuning techniques, such as adapters, LoRA, and prompt tuning.

---

> ### Author Response · Authors · 2023-11-11
> **Individual Response (1a)**
>
> Thank you for your constructive comments that make the paper solid. To begin, we would like to provide the following responses, along with a list of planned action items.
>
> ===
>
> > It would be beneficial if authors include visualizations of $\boldsymbol{v}\_{\boldsymbol{v}, \boldsymbol{\phi}}(x)$ and $\boldsymbol{v}\_{\boldsymbol{\theta}_0,\boldsymbol{\phi}}$ to provide an inituitive understanding of the distinctions between the original zero-shot model, the fine-tuned model with exisiting methods, and the fine-tuned model with the proposed emthod.
>
> We appreciate your insightful suggestion regarding visualizations, offering valuable insights to readers. As these vectors have high dimensions, such as M=80, we project them into __a two-dimensional space using t-SNE (Action Item #6)__. We have applied this approach to models fine-tuned through three methods -- FT, CAR-FT, and Lipsum-FT -- in the DomainNet scenario. The outcomes align with our expectations: FT shows the most significant distinction, followed by CAR-FT, and Lipsum-FT displays the least. We will include an additional section containing these results. Once again, thank you for your constructive comment!
>
> > Expanding the experiments to involve various domains as reference data for fine-tuning and using other domains for evaluation would enhance the comprehensiveness of the study. This approach can shed light on the adaptability and robustness of the proposed method in different real-world scenarios.
>
> Although we are currently exploring two scenarios that include DomainNet and ImageNet, it is clear that conducting additional experiments with varied choices of reference data will enhance the paper. Therefore, __we intend to showcase supplementary results using a different split as the reference data (Action Item #7).__ For instance, Trivedi et al. (2023) employed DomainNet-S as the reference data.
>
> > It remains unclear whether there exists a weight for incorporating the regularization term proposed in Eq. (7) into the loss function. In Appendix B.2, authors have discussed existing methods that involve weights related to the regularization term, such as $\lambda_{L2SP}$ in Eq. (10), $\lambda_{KD}$ in Eq. (12), and $\lambda_{CAR-FT}$ in Eq. (12). Authors have also detailed how to select these hyperparameters. However, there seems no mention of how the weight for the proposed method is determined.
>
> We appreciate your valuable feedback on the hyperparameter setup. In response, we adjusted the weight values within the hyperparameter space, aligning with the approach used for CAR-FT. To elaborate, we swept over {0.1, 0.2, 0.5, 1.0, 2.0, 5.0} and set 1.0 for both DomainNet and ImageNet scenarios. All hyperparameters for each method were adjusted to achieve the best performance on the reference data, as out-of-distribution data should not be considered in the fine-tuning procedure. We will clarify this point in the revised version of the paper.
>
> > The precision of the standard deviation values in Table 2(b) should be improved. The values of NLL are presented accurately to two decimal places, whereas the standard deviation values are limited to only one decimal place, with many of them showing as 0.0. Ensuring consistent precision in reporting these values would enhance the clarity and reliability of the results.
>
> Thanks for your thorough feedback on our experimental results! We inadvertently utilized the {:.1f} formatter for NLL, and __we will revise tables to maintain consistent precision when reporting numerical results (Action Item #8).__
>
> > There may be a typo in the gradient descent update rule presented in Eq. (9). It should be as follows: $\boldsymbol{\theta}\_t = \boldsymbol{\theta}\_{t-1} - \eta \nabla_{\boldsymbol{\theta}} L_{CE}(\boldsymbol{\theta})$. It's advisable for the authors to thoroughly review other equations to ensure they are accurately represented.
>
> Thanks for fixing the typo! We will double-check other formulas for errors as you mentioned.
>
> > It would be interesting to know if the proposed method is applicable to various fine-tuning techniques, such as adapters, LoRA, and prompt tuning.
>
> We believe our approach will effectively operate when features extracted from the vision model undergo changes for the same input. For instance, LoLA, where additional low-rank factors undoubtedly lead to changes in features extracted from the same input, will be compatible with the Lipsum-FT regularization. Conversely, for some prompt learning or adapter approaches, where features extracted from the same input remain unaltered, applying the Lipsum-FT regularization would not be straightforward.

---

> > ### Author Response · Authors · 2023-11-11
> > **Individual Reponse (1b)**
> >
> > ===
> >
> > __We will share the upcoming action items once they are completed. If there are any remaining concerns, please let us know. Otherwise, we would like to ask you to raise your assessment accordingly.__
> >
> > __Also, we kindly note that implementing and training for Action Item #7 requires extra resources and time. We will make every effort to share the results during the rebuttal period. However, even if the results are not ready by then, we will be dedicated to integrating them into the camera-ready version. We sincerely request the consideration of this matter for your assessment.__
> > * __(Action Item #6) A t-SNE visualizations of energy vectors before and after fine-tuning.__
> > * __(Action Item #7) A supplementary results using a different split as the reference data.__
> > * __(Action Item #8) Maintaining consistent precision in tables.__

---

> > > ### Comment · Reviewer_MuzV · 2023-11-20
> > >
> > > Authors' responses have addressed my response, and I have raised my rating to 6. I hope the authors could refine the changes and manuscript in their final version to make the paper easy to read.

---

> > > > ### Author Response · Authors · 2023-11-20
> > > >
> > > > We are pleased that our revisions have met your satisfaction! We will carefully address the matters raised during the discussion period in the final version of the manuscript. Thank you for your positive evaluation of our work!

---

### Author Response · Authors · 2023-11-12
**Global Response (1)**

We are pleased to present the first revision to all of you: [https://openreview.net/pdf?id=2JF8mJRJ7M](https://openreview.net/pdf?id=2JF8mJRJ7M)

The newly introduced material includes preliminary outcomes for each action item, and we intend to incorporate additional results in the final version of the paper, such as t-SNE plots for alternative baseline methods and ImageNet (Action Item #6). __Note that these materials are collected in Appendix C for immediate readability, and we will incorporate them into suitable sections in the firnal version of the paper__:
* See Appendix C.4 for (Action Item #1) An additional paragraph regarding energy-based models in the related work section.
* See Appendix C.5 for (Action Item #2) Improving readability of Figures 1 and 6.
* See Appendix C.3 for (Action Item #5) An additional analysis on training costs.
* See Appendix C.1 for (Action Item #6) A t-SNE visualizations of energy vectors before and after fine-tuning.
* See Appendix C.2 for (Action Item #7) A supplementary results using a different split as the reference data.
* Correct typos in Equations (9, 10, 12, 14) for the update rules (change plus to minus signs). Once again, we appreciate the detailed feedback from Reviewer MuzV!

===

Furthermore, we are presently engaged in the following activities:
* (Action Item #3) A conceptual figure demonstrating the overall idea of the work.
* (Action Item #4) Showing example text randomly generated for Lipsum-FT.

We are presently creating a conceptual diagram that represents these two elements, intending to assist readers in comprehending the overarching concepts outlined in the paper.

* (Action Item #8) Maintaining consistent precision in tables.

Due to the presence of standard deviation values at 0.00 for `{:.2f}` precision, we are contemplating adjusting it to `{:.3f}` precision.

===

__We are grateful for the constructive feedback once more, as it contributes to the robustness of our paper. If there are any remaining concerns, please let us know. Otherwise, we would like to ask you to raise your assessment accordingly.__

---

### Author Response · Authors · 2023-11-14
**Global Response (2)**

We are happy to present the second revision to all of you: https://openreview.net/pdf?id=2JF8mJRJ7M
* __(Action Item #3 and #4)__ Illustrated in Figure 17 is a conceptual representation showcasing the core concept of the study, along with randomly generated example text. We believe it improves readers' comprehension of the proposed methodology. Thanks again for valuable and constructive comments from Reviewer 26HN!
* __(Action Item #8)__ Table 11 presents a revised version of Table 2, evaluating predictive uncertainty using NLL and ECE. The {:.3f} formatter maintains consistency in presentation and improves clarity (please note that ECE values are no longer represented as percentages). We appreciate the thorough review provided by Reviewer MuzV once again!

===

Moreover, we would like to highlight the following four dimensions of our work: originality, quality, clarity, and significance.

* __Originality.__ Existing methods for robust fine-tuning typically aim to prevent the fine-tuned model from losing valuable knowledge acquired during pre-training. In our specific approach for pre-trained vision-language models, we highlight the significance of maintaining the zero-shot connections between vision and language models. Notably, our empirical findings indicate that quantifying these connections is feasible through the concept of energy, which can be easily computed by providing _random texts_ to the language model. This unique idea unquestionably introduces novelty to our methodology.

* __Quality and Clarity.__ Judging by the responses from reviewers, it seems that the paper clearly communicates (1) the problem statement concerning robust fine-tuning, (2) the interpretation based on the perspective of energy models, and (3) extensive experimental results demonstrating the effectiveness of the proposed approach. Furthermore, we believe the additional visual elements (Figure 15 from Action Item #6; Figure 17 from Action Items #3 and #4) included through revisions during the rebuttal period further improve the quality and clarity of the paper.

* __Significance.__ One of the key strengths of our methodology lies in its effectiveness despite its simplicity. Our suggested Lipsum-FT algorithm relies solely on the outputs of a pre-trained language model for _random texts_, eliminating the need for any intricate adjustments to either the input texts or the language model. At the same time, its simplicity marks our work as a pioneering effort addressing the language modeling aspect of vision-language models in robust fine-tuning. Given the expectation that future research will unveil more innovative strategies for utilizing language models, this work undoubtedly contributes significantly to the community.

===

Once again, we appreciate the valuable feedback provided by all reviewers, which makes the paper solid. __If there are any remaining concerns, please let us know. Otherwise, we kindly request that you adjust your assessment accordingly.__

---

### Author Response · Authors · 2023-11-16
**Global Response (3)**

We made the third revision with additional elements, such as t-SNE visualizations for other baselines in both DomainNet and ImageNet scenarios (Figure 15) and standard deviations from three measurements for experiments involving various domains as reference data (Table 10): https://openreview.net/pdf?id=2JF8mJRJ7M

We are thankful for the constructive feedback provided by the reviewers. Since the discussion period is entering its last week, we would like to know if there are any remaining concerns. Furthermore, we want to highlight that our revisions have effectively addressed the following primary issues raised by the reviewers:
* (Section C.1) Figure 15 depicts two-dimensional t-SNE visualizations of $v_{\theta,\phi}$ and $v_{\theta_{0},\phi}$ demonstrating our proposed Lipsum-FT displays the least distinction compared to all other baselines. As Reviewer MuzV mentioned, it provides an intuitive understanding of the distinctions between the original zero-shot and the fine-tuned models.
* (Section C.2) Table 10 validates that our proposed Lipsum-FT consistently surpasses baseline methods, even in experiments involving various domains as reference data for fine-tuning and using other domains for evaluation. As Reviewer MuzV mentioned, it sheds light on the adaptability and robustness of the proposed method.
* (Section C.3) It provides the wall-clock times for each fine-tuning method. As Reviewers 2e7P and 26HN suggested, it offers a clearer understanding of the computational complexity of our proposed Lipsum-FT approach.
* (Section C.5) Figure 17 depicts an overall idea of our proposed Lipsum-FT approach along with example texts randomly generated. As Reviewer 26HN suggested, we believe this illustration aids readers in understanding the concepts quickly.

If there are no remaining concerns, we sincerely request your positive reassessment of our work, as our revisions have effectively resolved all previously identified issues. Thanks once again for dedicating your time and effort to reviewing the paper.

Sincerely,
Authors of submission 6807

---

### Meta-Review · Area_Chair_vd47 · 2023-12-02

**Metareview:**

The paper proposes a novel robust fine-tuning algorithm, Lipsum-FT, that effectively utilizes the language modeling aspect of the vision-language pre-trained models. The proposed method is validated on distribution shift scenarios in DomainNet and ImageNet.
Pros:
* Results on DomainNet and ImageNet are impressive.
* A simple and effective regularization term.
Cons:
* limited novelty

The authors' rebuttal addresses most reviewers' concerns. Hence, three reviewers give rating 6.
Reviewer 2e7P would like the authors to include a comparison with more ZSL methods that focus on evaluating seen vs. unseen classes performance. The authors show some results trying to address the concern but received no more response from 2e7P.
AC thinks the focus of the paper is not ZSL method evaluating seen vs. unseen classes performance. The focus is on distribution shift scenarios under fixed classes. Hence, AC focuses more on the other three reviewers and suggests acceptance.

**Justification For Why Not Higher Score:**

limited novelty and, compared to general ZSL issue, the robus finetuning task has a smaller scope.

**Justification For Why Not Lower Score:**

Results on DomainNet and ImageNet are impressive.

---

### Decision · Program_Chairs · 2024-01-16

Accept (poster)